# Recovery of Lutacidiplasmatales archaeal order genomes suggests convergent evolution in Thermoplasmatota

Paul O. Sheridan [1,2], Yiyu Meng[1], Tom A. Williams [2] & Cécile Gubry-Rangin [1✉]

The Terrestrial Miscellaneous Euryarchaeota Group has been identified in various environments, and the single genome investigated thus far suggests that these archaea are anaerobic sulfite reducers. We assemble 35 new genomes from this group that, based on genome analysis, appear to possess aerobic and facultative anaerobic lifestyles and may oxidise rather than reduce sulfite. We propose naming this order (representing 16 genera) "Lutacidiplasmatales" due to their occurrence in various acidic environments and placement within the phylum Thermoplasmatota. Phylum-level analysis reveals that Thermoplasmatota evolution had been punctuated by several periods of high levels of novel gene family acquisition. Several essential metabolisms, such as aerobic respiration and acid tolerance, were likely acquired independently by divergent lineages through convergent evolution rather than inherited from a common ancestor. Ultimately, this study describes the terrestrially prevalent Lutacidiciplasmatales and highlights convergent evolution as an important driving force in the evolution of archaeal lineages.

[1] School of Biological Sciences, University of Aberdeen, Aberdeen, UK. [2] School of Biological Sciences, University of Bristol, Bristol, UK. ✉email: c.rangin@abdn.ac.uk

The role of archaea in soils is understudied, with many novel lineages performing previously unknown metabolic functions within the myriad of ecosystems they inhabit[1,2]. The Terrestrial Miscellaneous Euryarchaeota Group (TMEG) represent one of these enigmatic lineages. TMEG are members of the phylum Thermoplasmatota[3] (previously the superclass Diaforarchaea[4]), whose members occupy a wide variety of environments and ecological niches[5]. TMEG were initially described based on metagenomic clones from South African gold mines[6] and subsequently found to represent a distinct taxonomic order[7]. They inhabit numerous environments such as soil[8], acid mine drainage[7] and marine subsurface sediments[9]. Functional TMEG metabolism predictions are currently based on a single studied genome, TMEG-bg1, which was reassembled from a metagenome of deep anoxic peat layers[8]. Based on this analysis, these organisms were predicted to be anaerobes that degrade long-chain fatty acids and reduce sulfite as a terminal electron acceptor, producing hydrogen sulfide as a metabolic end product. However, it is unclear how well this single genome represents the metabolism of the order and how much metabolic diversity exists between its different lineages.

Several novel groups of Thermoplasmatota have recently been described from a myriad of environments[10–13]. These genomes provide the opportunity to study the complex evolutionary history of this group, which has been proposed to be a critical model of complex archaeal evolution and adaptation to contrasted environments[5]. This analysis aimed at investigating outstanding questions in the evolution of Thermoplasmatota, including whether aerobic respiration and acid tolerance in specific lineages were acquired either vertically from a common ancestor or laterally (either once or multiple times), and whether the last common ancestor of Thermoplasmatota was a thermophile whose descendants transitioned to mesophily on numerous occasions independently or vice versa.

We assembled 35 TMEG-related genomes from acidic soils and reclassified the group as a novel order Lutacidiplasmatales. In contrast to hypotheses based on analysis of the first available genome for the group, these organisms are predicted by genome analysis to be capable of aerobic respiration and oxidise, rather than reduce, sulfite generated from thiosulfate. Additionally, we reveal by systematic gene tree-species tree reconciliation and single gene tree analysis that essential metabolic genes, such as those involved in aerobic respiration, have been laterally acquired independently by multiple lineages during the evolution of Thermoplasmatota, rather than having been inherited vertically from a common ancestor. The addition of Lutacidiplasmatales to the analysis of Thermoplasmatota revealed the convergent evolution of genes involved in aerobic respiration, acid tolerance and glycolysis. Therefore, this work contributes to our understanding of niche adaptation in archaea, particularly in lineages with complex evolutionary histories involving multiple environmental transitions.

## Results

### Assembly and classification of terrestrial miscellaneous euryarchaeota group genomes

Nine surface (0–15 cm) and six subsurface (30–60 cm) acidic soils were sampled from 11 locations around Scotland (Supplementary Data 1). Environmental DNA was extracted from these samples and subjected to high-throughput sequencing. Reconstruction of metagenome-resolved genomes recovered 35 genomes related to the Terrestrial Miscellaneous Euryarchaeota Group (TMEG), based on GTDB relative evolutionary divergence scores. Altogether, we obtained 22 and 13 genome sequences from surface and subsurface soils, respectively (Table 1). A single genome (TMEG-bg1) has been previously investigated from subsurface anoxic peat layers[8], and another genome (UBA184[14]) has been released in GTDB. In addition, there is presently no TMEG cultured representative. The newly assembled TMEG genomes were of relatively high quality, with average completeness of 78% (range: 45–98%) and average contamination of 3% (range: 0–9%) (Table 1). These TMEG genomes appeared to be of relatively low abundance within their environments, with an average relative abundance of 0.4% (range: 0.1–1.4%), estimated by metagenomics sequence read recruitment (Supplementary Data 2). In addition, we also assembled a genome from the related order Lunaplasmatales[10].

These 35 new TMEG genomes and the two publicly available ones (TMEG-bg1 and UBA184) were affiliated with the same order and family, based on their relative evolutionary divergence (Supplementary Data 2). They were divided into 16 genera and 24 species based on their average amino acid identity (Supplementary Data 3). The 16S rRNA gene similarities agree with the genomes representing a single family with multiple genera (Supplementary Data 4). However, the recovery rate of 16S rRNA genes for these genomes was only 41% (15 of the 37 genomes).

We selected the genome AcS3-62 as the type material for classifying this novel order as this genome meets the quality criteria for type material suggested for MIMAGs[15,16]. Indeed, the AcS3-62 genome has high genome completeness (97% complete), has a low level of genome fragmentation (29 contigs), possesses the 5S, 16S and 23S rRNA genes, encodes for all 20 regular amino acids tRNAs and is a member of the most represented genus in our dataset (Supplementary Data 5).

The full-length 16S rRNA gene sequence query from AcS3-62 against the NCBI nr nucleotide database (www.ncbi.nlm.nih.gov) revealed sequences with family-level similarity (>90%) in a wide variety of environments, including forest, fen and peat soils, mine drainage, hot springs and rivers over a wide thermal range (10–78 °C) (Supplementary Data 6). Despite the large variety of habitats, they all appear to be strongly acidic (pH < 4.5), indicating that this order-level group occupies a wide range of acidic environments. The AcS3-62 16S rRNA gene was additionally queried against the extensive collection of 16S rRNA libraries in IMNGS[17], an integrated platform for analysing raw sequence read archives. Similar sequences (>90% similarity) were present in diverse environments but were particularly prevalent in acidic environments such as mine drainage and peat soils (32 and 31% of libraries, respectively) (Supplementary Fig. 1, Supplementary Data 7).

Hence, we propose the name Lutacidiplasmatales for this order, with 'Luti' and 'acidi' referring to its prevalence in acidic soil environments, and 'plasma' referring to its classification within the Thermoplasmatota.

### Metabolic traits of the Lutacidiplasmatales

In contrast to the previously investigated TMEG-bg1[8], the aerobic respiration terminal oxidase (Complex IV) was detected in almost all Lutacidiplasmatales presenting higher genome completeness (19 of 22 genomes), suggesting a common aerobic metabolism in Lutacidiplasmatales (Fig. 1; Supplementary Data 8). The complex IV genes are adjacent to genes encoding complex III of the electron transfer chain, biosynthesis of haem A and O, twin-arginine translocation system, cupredoxin and six other proteins of unknown function (with 5 of them possessing transmembrane domains) (Supplementary Data 9). In addition, the microaerobic respiration terminal oxidase, cytochrome bd ubiquinol oxidase genes cydA and cydB[18–20] were only present in two closely related genomes (TMEG-bg1 and UBA184), suggesting adaptation of these organisms to environments where molecular oxygen is scarce. The cytochrome bd ubiquinol oxidases identified in this

**Table 1 Genome characteristics of newly sequenced metagenome-assembled genomes.**

| | Completeness | Contamination | GC% | Adjusted genome size (bp) | Adjusted # CDS | Contig # | Proteome novelty (%) | 5S | 16S | 23S | Ecosystem |
|---|---|---|---|---|---|---|---|---|---|---|---|
| *Ca.* Lutacidiplasmatales | | | | | | | | | | | |
| AcS10-2 | 48 | 0 | 70 | 1.8E + 06 | 1984 | 263 | 69 | 0 | 0 | 0 | Pine forest |
| AcS11-139 | 58 | 3 | 71 | 1.8E + 06 | 2234 | 591 | 75 | 1 | 0 | 0 | Pine forest |
| AcS11-62 | 95 | 2 | 67 | 2.3E + 06 | 2062 | 129 | 65 | 1 | 2 | 0 | Pine forest |
| AcS1-3 | 59 | 0 | 68 | 3.2E + 06 | 3238 | 219 | 64 | 1 | 0 | 0 | Pine forest |
| AcS13-54 | 93 | 2 | 69 | 2.2E + 06 | 1027 | 17 | 62 | 1 | 1 | 2 | Pine forest |
| AcS1-36 | 96 | 4 | 69 | 9.7E + 05 | 1861 | 28 | 56 | 1 | 0 | 0 | Pine forest |
| AcS14-35 | 55 | 2 | 70 | 1.8E + 06 | 1915 | 213 | 67 | 1 | 1 | 0 | Marsh |
| AcS1-45 | 83 | 2 | 67 | 2.3E + 06 | 2259 | 222 | 66 | 1 | 0 | 0 | Pine forest |
| AcS1-67 | 84 | 2 | 67 | 1.9E + 06 | 1850 | 37 | 62 | 1 | 0 | 0 | Pine forest |
| AcS2-54 | 55 | 9 | 71 | 1.9E + 06 | 1945 | 139 | 64 | 1 | 0 | 0 | Pine forest |
| AcS3-62 | 97 | 2 | 70 | 2.2E + 06 | 2044 | 29 | 63 | 1 | 1 | 1 | Pine forest |
| AcS3-69 | 54 | 4 | 69 | 2.0E + 06 | 2277 | 411 | 74 | 0 | 0 | 0 | Pine forest |
| AcS4-12 | 45 | 3 | 69 | 2.0E + 06 | 2029 | 216 | 65 | 1 | 0 | 0 | Birch forest |
| AcS4-16 | 77 | 9 | 70 | 2.2E + 06 | 2218 | 356 | 69 | 1 | 1 | 0 | Birch forest |
| AcS4-93 | 94 | 7 | 68 | 2.4E + 06 | 2272 | 205 | 67 | 1 | 0 | 0 | Birch forest |
| AcS5-107 | 98 | 2 | 68 | 2.5E + 06 | 2284 | 52 | 66 | 1 | 0 | 0 | Pine forest |
| AcS5-109 | 58 | 6 | 67 | 2.4E + 06 | 2707 | 400 | 75 | 0 | 0 | 0 | Pine forest |
| AcS5-116 | 98 | 3 | 68 | 2.2E + 06 | 1982 | 51 | 63 | 1 | 0 | 0 | Pine forest |
| AcS5-34 | 92 | 2 | 68 | 2.1E+06 | 2023 | 146 | 65 | 1 | 0 | 0 | Pine forest |
| AcS5-58 | 97 | 2 | 69 | 2.2E + 06 | 2037 | 27 | 65 | 1 | 0 | 1 | Pine forest |
| AcS5-77 | 74 | 5 | 69 | 2.3E + 06 | 2426 | 309 | 69 | 0 | 0 | 0 | Pine forest |
| AcS5-86 | 46 | 4 | 70 | 2.5E + 06 | 3014 | 411 | 76 | 0 | 0 | 0 | Pine forest |
| SubAcS10-22 | 95 | 2 | 68 | 1.6E + 06 | 1492 | 5 | 55 | 1 | 1 | 1 | Pine forest |
| SubAcS10-31 | 94 | 3 | 69 | 1.9E + 06 | 1863 | 105 | 63 | 1 | 1 | 1 | Pine forest |
| SubAcS11-52 | 89 | 2 | 68 | 1.6E + 06 | 1438 | 6 | 54 | 1 | 1 | 0 | Pine forest |
| SubAcS13-24 | 94 | 1 | 69 | 1.7E + 06 | 1655 | 95 | 60 | 1 | 1 | 1 | Pine forest |
| SubAcS13-35 | 65 | 1 | 69 | 1.7E + 06 | 1723 | 164 | 64 | 1 | 0 | 0 | Pine forest |
| SubAcS13-64 | 93 | 3 | 68 | 1.9E + 06 | 1802 | 97 | 61 | 1 | 1 | 0 | Pine forest |
| SubAcS14-22 | 51 | 6 | 70 | 3.0E + 06 | 3302 | 569 | 75 | 1 | 0 | 0 | Marsh |
| SubAcS15-120 | 92 | 2 | 69 | 2.2E + 06 | 2076 | 106 | 65 | 1 | 0 | 1 | Grassland |
| SubAcS15-25 | 88 | 3 | 68 | 1.4E + 06 | 1454 | 161 | 59 | 1 | 1 | 0 | Grassland |
| SubAcS15-31 | 96 | 2 | 66 | 2.2E + 06 | 2111 | 5 | 62 | 1 | 1 | 1 | Grassland |
| SubAcS15-47 | 56 | 2 | 65 | 2.6E + 06 | 2798 | 373 | 70 | 1 | 0 | 1 | Grassland |
| SubAcS15-67 | 93 | 3 | 68 | 2.0E + 06 | 1902 | 159 | 62 | 1 | 1 | 1 | Grassland |
| SubAcS9-61 | 89 | 4 | 69 | 2.0E + 06 | 1981 | 311 | 66 | 1 | 1 | 0 | Peaty gley |
| *Ca.* Lunaplasmatales | | | | | | | | | | | |
| SubAcS15-131 | 60 | 0 | 65 | 1.3E + 06 | 1633 | 301 | 64 | 0 | 0 | 0 | Grassland |

Genome size and CDS number were adjusted for completeness. The percentage completeness and contamination of each genome is presented. The copy number of 5S, 16S and 23S rRNA genes are indicated for each genome under the headings 5S, 16S and 23S. More detailed information on all genomes used in study can be found in Supplementary Data 2.

study are members of the rarer quinol:O$_2$ oxidoreductase, qOR3 family, based on the cydA subfamily database[21]. Several Lutacidiplasmatales genomes also contain the aldehyde dehydrogenase (ALDH) and aldehyde-alcohol dehydrogenase (*adh*E), which are responsible for producing acetate and ethanol, respectively. The presence of these genes suggests that these organisms may also be facultative anaerobes.

Lutacidiplasmatales may be obligate heterotrophs. Indeed, a common feature among Lutacidiplasmatales genomes is the absence of complete carbon fixation pathways, including the dicarboxylate-hydroxybutyrate, the hydroxypropionate-hydroxy-butyrate, the reductive acetyl-CoA, the reductive hexulose-phosphate or the Wood-Ljungdahl pathways (Supplementary Data 8). Lutacidiplasmatales genomes also lack methanogenesis genes. Additionally, many Lutacidiplasmatales genomes possess a complete glycolytic pathway, notably having the limiting 6-phosphofructokinase gene. Most genomes contain multiple copies of the *glo*A gene, which detoxifies the glycolysis by-product methylglyoxal, suggesting glycolysis as an essential pathway in those genomes. A complete propionate to succinate pathway is also present and includes the marker genes propionyl-CoA carboxylase beta chain *pcc*B, methylmalonate-semialdehyde dehydrogenase *mms*A and methylmalonyl-CoA/ethylmalonyl-CoA epimerase MCEE, suggesting that propionate could be another carbon and energy source under aerobic environmental conditions.

Degradation of LCFA via β-oxidation was previously proposed for TMEG-bg1 genome[8]. This function appears to be a common feature in Lutacidiplasmatales, with most genomes possessing the necessary genes for utilising long/medium-chain fatty acids (Fig. 1). Lutacidiplasmatales genomes also contain multiple alpha-glucan degrading glycoside hydrolase (GH) genes involved in the degradation of carbohydrate storage, such as amylose (GH57), trehalose (GH15) and glycogen (GH133) (Supplementary Data 10). Extracellular peptidases were also detected, with the S53 family, thermopsin and A4 family peptidases being the most prevalent (Supplementary Data 11). Notably, these peptidases are active at low pH[22–24], and extracellular S53 and thermopsin peptidases are present in the acidophilic Thermoplasmatales species *Cuniculiplasma divulgatum*[25]. Lutacidiplasmatales also possess the peptide/nickel ABC transport system and genes for the degradation of amino acids, such as histidine, glutamate, glutamine, glycine, valine, leucine and isoleucine (Supplementary Data 8).

Lutacidiplasmatales have an acidophilic lifestyle, as demonstrated by multiple genetic pathways. They encode an *arc*A arginine deiminase, which was demonstrated as an important gene for acid tolerance in other organisms[26–28]. They also possess the acid-tolerant form of the V/A- ATPase (see section Results: Origination and evolution of critical metabolisms in Thermoplasmatota). This clade of V-type-like ATPase is adapted to function under highly acidic conditions and has been laterally acquired by many distantly related acidophilic archaea[29]. Other mechanisms also support the acidophilic specialisation of Lutacidiplasmatales. In particular, most Lutacidiplasmatales possess the *Kdp* potassium transporter (EC:3.6.3.12), which is involved in pH homeostasis in acidophiles by generating reverse membrane potential[30,31]. Lutacidiplasmatales genomes also encode other stress resistance genes such as Uvr excinuclease, involved in DNA repair from ultraviolet DNA damage[32], the *Mut*LS endonuclease, mitigating the mismatch repair during DNA replication, and the nickel-containing superoxide dismutase, conferring an efficient oxidative stress resistance in environments with high-nickel concentrations[33].

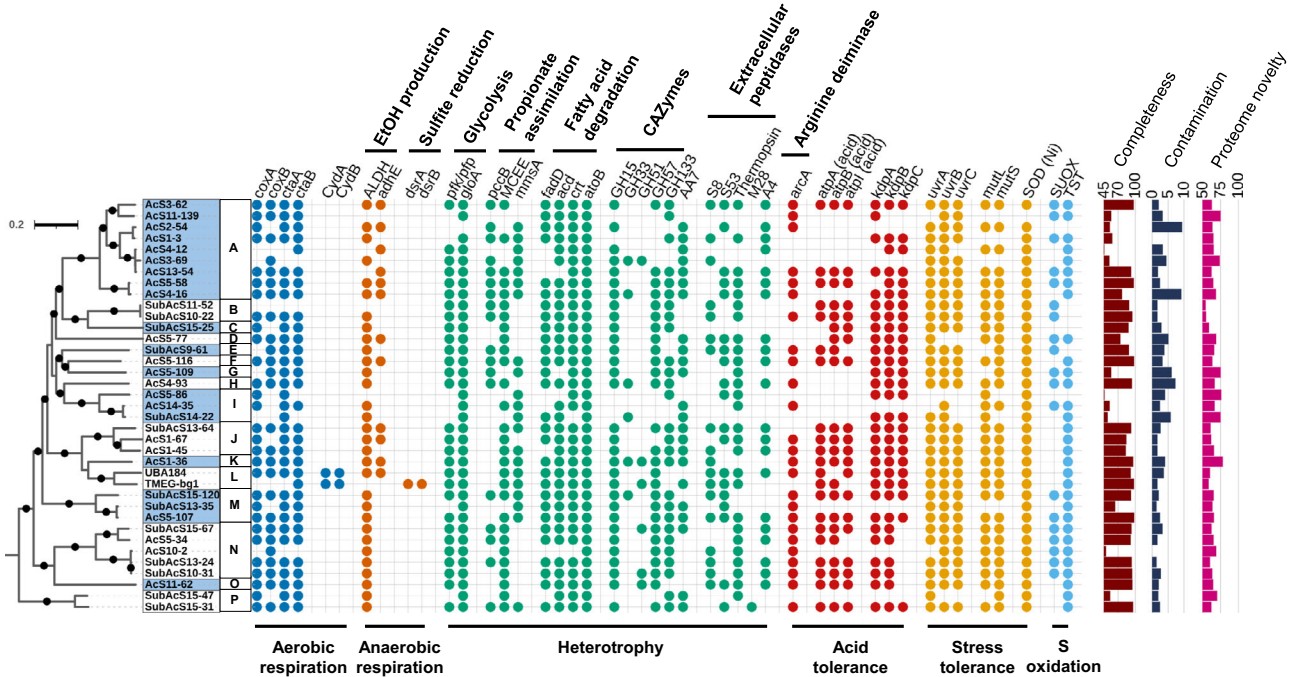

**Fig. 1 Phylogenomic tree of Lutacidiplasmatales and associated metabolism.** This tree includes 37 Lutacidiplasmatales genomes assembled from surface and subsurface soils (35 genomes obtained in this study and two published sequences). The tree was inferred by maximum likelihood from 738 concatenated phylogenetic markers, which were aligned separately and analysed using the best-fitting model for each alignment. The tree was rooted with the Methanomassiliicoccales strains *Methanoplasma termitum* MpT1, Methanomethylophilus archaeon BRNA1 and *Methanomethylophilus alvus* Mx1201. Dots indicate branches with >95% UFBoot and SH-aLRT support. Intermittent blue and white colours on the leaf labels indicate the 16 known genera of Lutacidiplasmatales, as determined by an amino acid identity of 70% or greater. *cox*A (haem-copper oxygen reductases, subunit A; K02274), *cox*B (haem-copper oxygen reductases, subunit B; K02275), *cta*A (haem a synthase; K02259), *cta*B (haem o synthase; K02257), *cyd*A (cytochrome bd ubiquinol oxidase, subunit A; K00425), *cyd*B (cytochrome bd ubiquinol oxidase, subunit B; K00426), *adh*E (alcohol dehydrogenase; K0407), *dsr*AB (dissimilatory sulfite reductase, subunits A and B; K11180 and K11181), *pfk/pfp* (ATP-dependent phosphofructokinase; K21071), *glo*A (lactoylglutathione lyase; K01759), *pcc*B (propionyl-CoA carboxylase, beta chain, K01966), MCEE (methylmalonyl-CoA/ethylmalonyl-CoA epimerase; K05606), *mms*A (methylmalonate-semialdehyde dehydrogenase; K00140), *fad*D (long-chain acyl-CoA synthetase; K01897), *acd* (acyl-CoA dehydrogenase; K00249), *crt* (enoyl-CoA hydratase; K01715), *fad*B (3-hydroxybutyryl-CoA dehydrogenase; K00074), *fad*A (acetyl-CoA acyltransferase; K00632), *ato*B (acetyl-CoA C-acetyltransferase; K00626), GH# (glycoside hydrolase family #; dbCAN), Pen amidase (Penicillin amidase), *arc*A (arginine deiminase; K01478), *atp*ABI (acid) (V/A-type atpase A, B and I subunits; K02117, K02118 and K02123, plus gene tree analysis), *kdp*ABC (K + transporting ATPase subunits A, B and C; K01546, K01547 and K01548), *uvr*ABC (excinuclease subunits A, B and C; K03701, K03702 and K03703), *mut*LS (DNA mismatch repair proteins L and S; K03572 and K03555), SOD (Ni) (nickel superoxide dismutase; K00518), SUOX (sulfite oxidase; PF00174, plus gene tree analysis) and TST (thiosulfate/3-mercaptopyruvate sulfurtransferase; K01011). Similar metabolic analysis expanded to the whole Thermoplasmatota genome database is presented in Supplementary Fig. 8. The predicted completeness, contamination and proteome novelty of each genome are indicated in the bar charts on the far right of the figure.

The sulfite reduction genes, *dsr*AB, of TMEG-bg1 were not detected in other Lutacidiplasmatales genomes. In contrast, most Lutacidiplasmatales genomes possess genes homologous to putative archaeal sulfite oxidases previously identified in Thaumarchaeota[34]. These genes may be involved in converting sulfite into sulfate, allowing the generation of ATP in oxidative phosphorylation. A phylogenetic tree of these genes in various prokaryotes and eukaryotes was reconstructed and was rooted at the position with the lowest ancestor deviation (see SI MAD rooting; Supplementary Fig. 2). This indicates that these archaeal enzymes form the closest prokaryotic sister clade to eukaryotic sulfite oxidases and nitrate reductases. Instead of the transmembrane domains found in Thaumarchaeota sulfite oxidases, the N-terminal region of the Lutacidiplasmatales sulfite oxidases possess an intrinsic disorder (i.e. natively unfolded) region (Supplementary Fig. 3).

Lutacidiplasmatales are predicted to be non-motile due to the lack of archaellum genes, including those present in closely related Thermoplasmatota, such as Thermoplasmatales or Poseidoniales[12,35]. Members of the Lutacidiplasmatales possess a high level of genome novelty (average 65%; range 54 – 76%)

(Fig. 1, Table 1, Supplementary Data 2). For example, the near-complete genome sequences of AcS3-62 encode 2,044 predicted proteins, but only 37% of these genes possessed a close homologue in the arCOG database[36] and 58% were assigned to ortholog groups in the KEGG database[37]. Given the high level of genome novelty, it is likely that our functional predictions underestimate the physiological repertoire of these organisms.

**Phylogenomic tree and genome characteristics of Thermoplasmatota.** Published phylogenies of Thermoplasmatota differ with respect to the placement of certain clades, including Poseidoniales and Thermoprofundales[5,10,13,38]. As a robust species phylogeny is essential in accurately predicting gene content change, we performed a range of phylogenomic analyses using different taxon sets (an expanded 124 genome set including all 35 newly sequenced Lutacidiplasmatales genomes, and a 100-genome subsample of the higher-quality genomes including 21 newly sequenced genomes). We used two sets of conserved single-copy marker gene and a set of conserved single-copy ribosomal genes. The trees were reconstructed using four

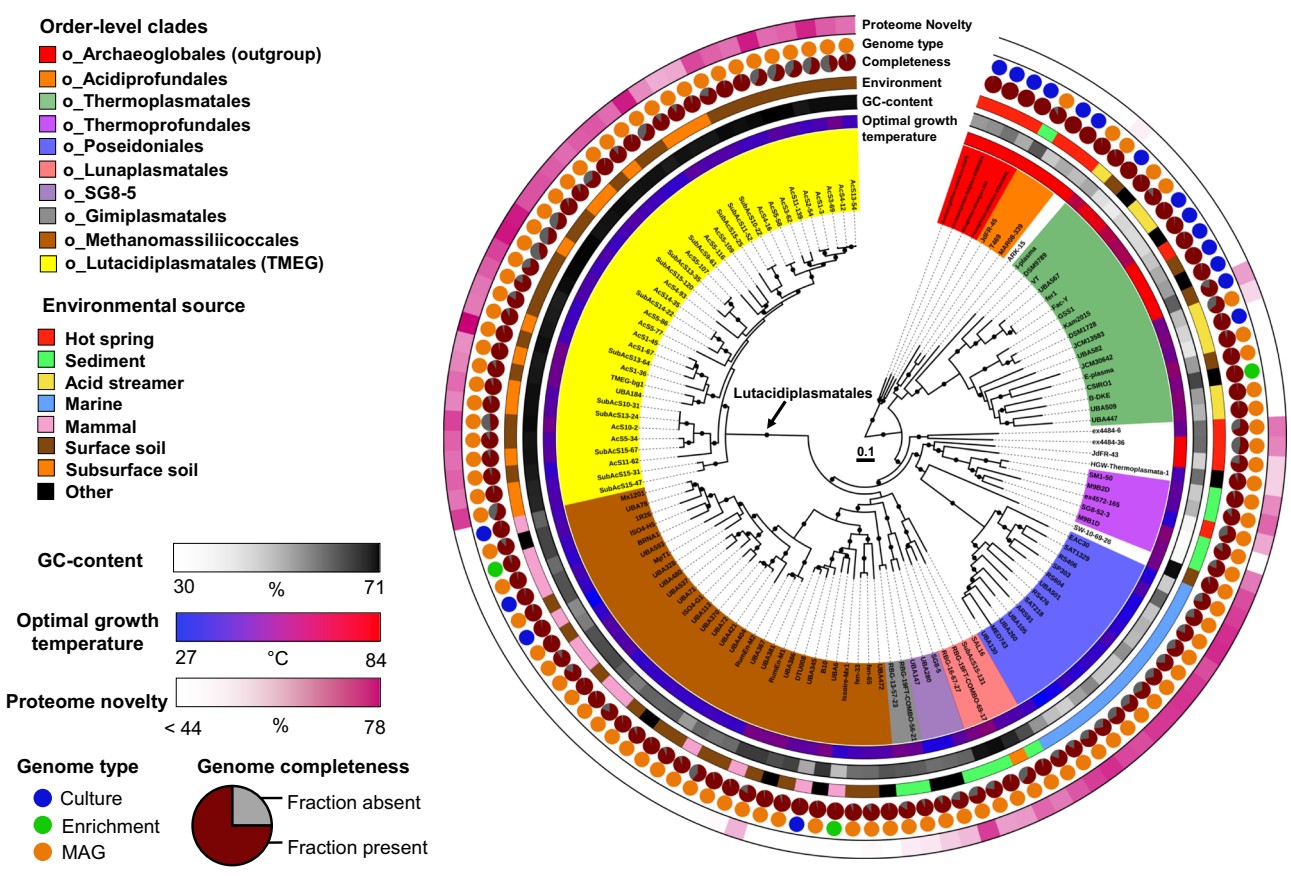

**Fig. 2 Phylogenomic tree of Thermoplasmatota.** This tree includes 120 Thermoplasmatota genomes from a variety of environments. The tree was inferred by maximum likelihood from 108 concatenated marker genes, aligned separately and analysed using the best-fitting model for each alignment. The tree was rooted with four Archaeoglobales genomes. Dots indicate branches with >95% UFBoot and SH-aLRT support. As previously published studies have recovered different phylogenies for Thermoplasmatota, species trees inferred using a variety of approaches were compared. This phylogeny was the most likely, based on approximately unbiased testing and this topology being reconstructed in six of the seven approaches applied here (see SI Phylogenomics). Protein novelty is defined as the percentage of encoded proteins that lack a close homologue in the arCOG database. Detailed genome information is given in Supplementary Data 1.

analytical approaches. More details are provided in SI Phylogenomics Methods and Results, Supplementary Fig. 4; Supplementary Data 12). The best phylogeny derived from this analysis is presented in Fig. 2. This phylogenetic topology was retrieved in 6 of the 8 tree reconstruction approaches tested and largely agrees with the 16S rRNA phylogenetic tree (Supplementary Fig. 5). The Lutacidiplasmatales order forms a monophyletic clade with the methanogenic Methanomassiliicoccales[39] and the recently described order-level groups Gimiplasmatales[13], SG8-5[40] and Lunaplasmatales[10] (Fig. 2).

The resolved phylogenomic tree allowed investigation of the genome dynamics in the whole phylum. Ancestral temperature preference was predicted by combining the in silico prediction of optimal growth temperature of extant Thermoplasmatota genomes with the reconstructed phylogenomic tree. It indicated that the last common ancestor of Thermoplasmatota was a thermophile, which transitioned to moderate temperature environments on several occasions independently during the subsequent diversification of Thermoplasmatota (Supplementary Fig. 6).

The Thermoplasmatota genome size is relatively consistent across the phylum, with an average of 1,954 protein-coding genes per genome (after adjustment for the completeness) [range 1,027–4,052] (Table 1). In contrast to Thaumarchaeota[34] and Cyanobacteria[41], genome expansion did not occur during the transition of Thermoplasmatota to terrestrial environments (Supplementary Fig. 7). An increase in GC content also happened

during the early evolutionary history of the Lutacidiplasmatales, Methanomassiliicoccales, SG8-5 order, Gimiplasmatales and Lunaplasmatales (Supplementary Fig. 7).

**Diversification and mechanisms of genome evolution in Thermoplasmatota.** We next investigated mechanisms of gene content evolution on the consensus Thermoplasmatota phylogeny. A selection of 96 higher-quality (>70% completeness, <5% contamination) Thermoplasmatota genomes plus four outgroup genomes from the Archaeoglobales were used to infer the evolutionary history of 6,050 gene trees throughout Thermoplasmatota evolution. The probabilistic ancestral genome reconstructions for every branch of the Thermoplasmatota phylogeny enabled the characterisation and quantification of gene content changes across their history (Supplementary Data 13). The majority of the 21,183 gene content gains in Thermoplasmatota occurred through 13,025 intra-phylum gene transfers (intra-LGT) (61% of gains), 4,947 originations (including inter-phyla gene transfers and de novo gene formation) (23% of gains) and 3,211 duplications of existing genes (15% of gains) (Fig. 3A). There were also 16,374 gene losses predicted (Fig. 3A). Therefore, gene transfer within the Thermoplasmatota phylum and gene loss were the two main mechanisms of gene content change in Thermoplasmatota evolution.

The distribution of the gene gains and losses across the Thermoplasmatota evolutionary history was determined by

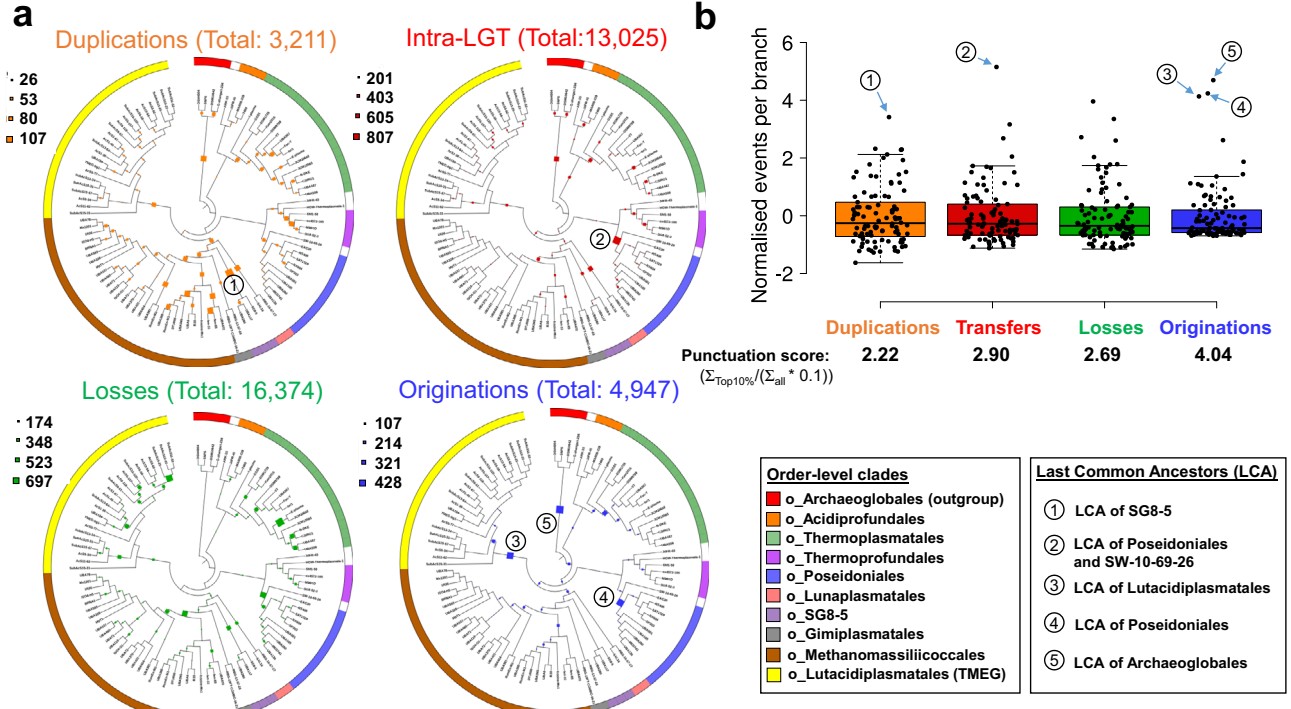

**Fig. 3 Inferred gene originations, duplications, intra-phylum transfers and losses during the evolution of Thermoplasmatota. a** The quantitative and qualitative predictions of the genome content changes were estimated across the Thermoplasmatota history and reported on the circular cladograms possessing the topology of the Thermoplasmatota tree (see Supplementary Fig. 6). The mechanisms of gene content changes quantified were duplication (copying of a gene within a genome), loss (loss of a gene within a genome), intra-phylum transfers (defined as the acquisition of a gene from another member of the same phylum) (Intra-LGT) and origination (defined as the acquisition of a gene from members of phyla outside the sampled genome set or by de novo gene formation). Scale numbers indicate the range of the predicted number of events for each given mechanism. The order-level classification of the genomes is indicated by the coloured bar surrounding the circle. **b** The boxplots represent normalised events per branch ((events per branch - $\mu$)/$\sigma$) for each mechanism. Numbered circles mark branches of the tree with the highest numbers of events, and the position of these branches is also indicated with numbered circles on the trees (in **a**). Horizontal lines within boxes indicate the medians, box boundaries indicate the 1st and 3rd quartiles, whiskers indicate the minima and maxima, and points beyond these whiskers are outliers. A punctuation score is measured for each given mechanism. It represents the sum of events in the 10% of branches with the highest event numbers divided by 10% of the sum of events into all branches ($\Sigma_{\text{events in top10\%}}/(\Sigma_{\text{events in all branches}} * 0.1)$).

estimating a punctuation score index for each of the four gene content change mechanisms (Fig. 3b). A higher punctuation score represented a lower dispersion of events across the phylogeny, with most events happening on a few branches of the phylogenomic tree. Gene family originations was the most punctuated mechanism of gene content change in Thermoplasmatota, with a punctuation score (PS) of 4.04, followed by intra-LGT (PS = 2.90), losses (PS = 2.69) and duplications (PS = 2.22) (Fig. 3b). Indeed, 16% of all gene family originations in Thermoplasmatota were predicted to have occurred in the last common ancestors (LCA) of Lutacidiplasmatales (383 originations) and Poseidoniales (392 originations) (Fig. 3; Supplementary Data 13). These originating gene families possessed a high degree of novelty, with only 22 and 8% having close homologues in the arCOG database for Lutacidiplasmatales and Poseidoniales, respectively (Supplementary Data 14). These originations were not followed by high duplication and loss rates in the subsequent lineages (see SI Gene duplication), as observed for some Thaumarchaeota lineages[34]. In contrast, gene duplication appears as a more important mechanism for the evolution of the Methanomassiliicoccales or the Thermoplasmatales (15 and 21% of the gene gains throughout these lineages, respectively). As gene families with less than four sequences were not included in gene tree-species tree reconciliation, the relatively low number of extant genomes in the order-level clades Acidiprofundales, Thermoprofundales,

Lunaplasmatales, Gimiplasmatales and SG8-5 may lead to underestimations of the number of originations for their LCAs.

**Origination and evolution of key metabolisms in Thermoplasmatota.** Following the probabilistic ancestral gene content reconstruction, we examined progressive gain and loss of genes throughout the evolutionary history of Thermoplasmatota by comparing the set of functional genes predicted in each parent ancestor reconstruction to their descendants (expanded description in Supplementary Fig. 8). We complemented these broad-scale systematic analyses with targeted phylogenetics of the genes underpinning key metabolic pathways, including homologues from other archaeal and bacterial genomes (Supplementary Data 15). We used approximate unbiased testing and other statistical methods to test the likelihood of single acquisitions into the phylum (Supplementary Data 16). This approach enables the prediction of gene family acquisitions even when the same gene family has been laterally acquired multiple times into the phylum. We focused on specific metabolic traits such as oxidative phosphorylation, acid tolerance, autotrophy and heterotrophy (Fig. 4).

*Oxidative phosphorylation.* Genes indicating aerobic respiration via complex IV were present in Lutacidiplasmatales, Thermoplasmatales, Poseidoniales and Lunaplasmatales (Supplementary

Data 8, Supplementary Fig. 9). The gene arrangement of the complex IV locus is highly variable between orders (Fig. 5a). Still, it is conserved within each order, except for the Thermoplasmatales genera *Ferroplasma*, *Acidiplasma* and *Picrophilus*, which possess a divergent *cox*AC-encoding complex IV locus. Genes encoding cytochrome c cytochromes (PF00034 or PF01322) were detected in the Archaeoglobales genomes *Archaeoglobus fulgidus* DSM4304, *Archaeoglobus veneficus* SNP6 and *Ferroglobus placidus* DSM10642, but were absent from all Thermoplasmatota genomes. The complex IV subunits in Lutacidiplasmatales, Thermoplasmatales and Lunaplasmatales are adjacent to blue copper proteins (PF06525, PF00127). These BCP redox proteins shuttle electrons from an electron donor to an electron acceptor[42] and have been proposed as an alternative electron carrier in the respiratory chain of the Thermoplasmatales species *Cuniculiplasma divulgatum*[25] and some other acidophiles[7,43].

The high gene arrangement divergence between orders suggests that the individual genes of complex IV may have markedly different evolutionary histories. Therefore, the phylogenies of the subunits *cox*A and *cox*B and the biogenesis genes *cta*A and *cta*B were compared with homologues from an expanded inter-domain set of prokaryotic genomes (Supplementary Data 15; Fig. 5b and Supplementary Fig. 10–13). The ancestral gene content reconstruction indicates that this trait (characterised by the marker gene *cox*A) was gained by Thermoplasmatota lineages on at least three independent occasions (Fig. 4). Targeted gene tree analysis, including genes from other archaea and bacteria, revealed that the *cox*A genes of Poseidoniales were highly divergent from those of Thermoplasmatales, Lutacidiplasmatales and Lunaplasmatales, clustering more closely to a clade of bacterial genes (Fig. 5b and Supplementary Fig. 10). This indicates that there were at least two independent acquisitions of *cox*A into Thermoplasmatota, and a single acquisition of *cox*A into Thermoplasmatota was rejected by an approximately unbiased (AU) test ($P < 0.01$). Despite the interdomain *cox*A divergence, all Thermoplasmatota *cox*A genes were members of the D- and K-channel possessing A1 subfamily of haem-copper oxygen reductases[44]. Multiple acquisitions during evolutionary history were also predicted for *cox*B, *cta*A and *cta*B genes (see SI Complex IV evolution).

The microaerophilic oxygen reductase cytochrome bd ubiquinol oxidase is present in most members of the Thermoplasmatales and in the Lutacidiplasmatales genomes TMEG-bg1 and UBA184 (Supplementary Fig. 9). Ancestral gene content reconstruction indicates that the cytochrome bd ubiquinol oxidase subunit *cyd*A originated in the LCA of Thermoplasmatales and has undergone two duplications in its descendants (Supplementary Fig. 14, Supplementary Data 17). The *cyd*A present in the two Lutacidiplasmatales members was likely transferred to their LCA from a member of the Thermoplasmatales order (0.9 TPP (posterior probability of transfers)) (Supplementary Data 17).

*Acid tolerance*. The energy-yielding V/A-type $H^+/Na^+$-transporting ATPases are present in all orders of the Thermoplasmatota (Supplementary Data 8). The V/A-type ATPase of most of the Thermoplasmatota orders possesses the H/GIKEC-FABD gene arrangement of the previously described halophilic V-type ATPase[29] (Fig. 6A), with Acidiprofundales differing only by subunit D being encoded in a different part of the genome than the rest of the ATPase complex. The Thermoplasmatales possess the KECFABDH/GI arrangement of the acidophilic V-type ATPase[29]. In contrast, the Lutacidiplasmatales possess the gene arrangement FABDH/GICEK, which has not been previously identified in archaea to the best of our knowledge.

A phylogenetic analysis of a concatenation of the three largest subunits of this complex (subunits A, B and I), including genes from other archaea and bacteria, revealed that a single acquisition

into the phylum could be statistically rejected ($P < 0.01$). The tree showed that the ATPases of all Thermoplasmatales and Lutacidiplasmatales genomes clustered with the acidophilic V-type ATPases[29] (Fig. 6B and Supplementary Fig. 15). In addition, the V/A-ATPases of Acidiprofundales, Thermoprofundales and Thermoplasmatota archaeon JdFR-43 were likely acquired independently of the other ATPases in Thermoplasmatota. Their clustering with other archaea present in deep-sea hydrothermal sediments probably reflects adaptation to conditions specific to this environment (such as pressure), as observed for the acid-tolerant V-type-like ATPases. Individual gene trees of the three subunits indicated the same clustering as the concatenated approach (Supplementary Fig. 16–18).

The acid tolerance associated arginine deiminase, *arc*A, gene, is present in most genomes of the Thermoplasmatales and Lutacidiplasmatales (Supplementary Fig. 9). It was predicted by ancestral gene content reconstruction to have been gained on two independent occasions into the LCAs of these orders. However, there is some evidence that the *arc*A present in the Thermoplasmatales LCA may have been acquired from another Thermoplasmatota, such as the clustering of all Thermoplasmatota genes together to the exclusion of bacterial homologues (Supplementary Fig. 19) and the indication of an *arc*A intra-phylum transfer into the Thermoplasmatales LCA by gene tree-species tree reconciliation (TPP 0.58) (Supplementary Data 18). However, an intra-LGT from Lutacidiplasmatales (the most likely donor) was poorly supported (TPP 0.27), so two independent acquisitions remain the most likely scenario.

*Autotrophy and heterotrophy*. Genes indicative of autotrophy were detected in some (but not all) lineages of Thermoplasmatota, and it is unclear whether Thermoplasmatota transitioned from a heterotrophic ancestor to autotrophy in some lineages or vice versa. The key genes acetyl-CoA decarbonylase, subunits gamma and delta (*cdh*ED) in the $CO_2$ assimilating Wood-Ljungdahl pathway were present in genomes of the Gimiplasmatales and Methanomassiliicoccales but absent from other members of the phylum. The ancestral gene content reconstruction suggested that these genes were acquired once in the LCA of these two orders (Fig. 4). This scenario is supported by an expanded phylogenetic analysis of these genes, including genes from other archaea and bacteria (Supplementary Figs. 20 and 21). The key methanogenesis genes, *mcr*AB, were only detected in the Methanomassiliicoccales and are predicted to have been acquired by the LCA of that order (Fig. 4).

The $CO_2$ assimilating RuBisCO system key gene, *rbc*L, was present in the genomes of the Acidiprofundales, SG8-5 order, and Methanomassiliicoccales and ancestral gene content reconstruction suggests that this gene was gained independently in the LCAs of these orders (Fig. 4). Phylogenetic analysis of this gene also supports at least three independent acquisitions, with Methanomassiliicoccales genes forming a divergent cluster and SG8-5 order genes clustering more closely with archaea from other phyla than with the Acidiprofundales genes (Supplementary Fig. 22). Additionally, a single acquisition of *rbc*L into the phylum was statistically rejected ($P < 0.01$).

The autotrophic carbon assimilating pathways, 3-hydroxypropionate bicycle, 3-hydroxypropionate/4-hydroxybutyrate cycle, reverse TCA and methylamine-formate reaction, were not detected in any members of the Thermoplasmatota or their ancestors (Supplementary Data 8).

The key glycolytic gene ATP-dependent phosphofructokinase *pfk* was present in most Thermoprofundales and Lutacidiplasmatales genomes but absent from most other genomes of the phylum (Supplementary Fig. 9, Supplementary Data 8). The ancestral gene content reconstruction predicted that this gene was

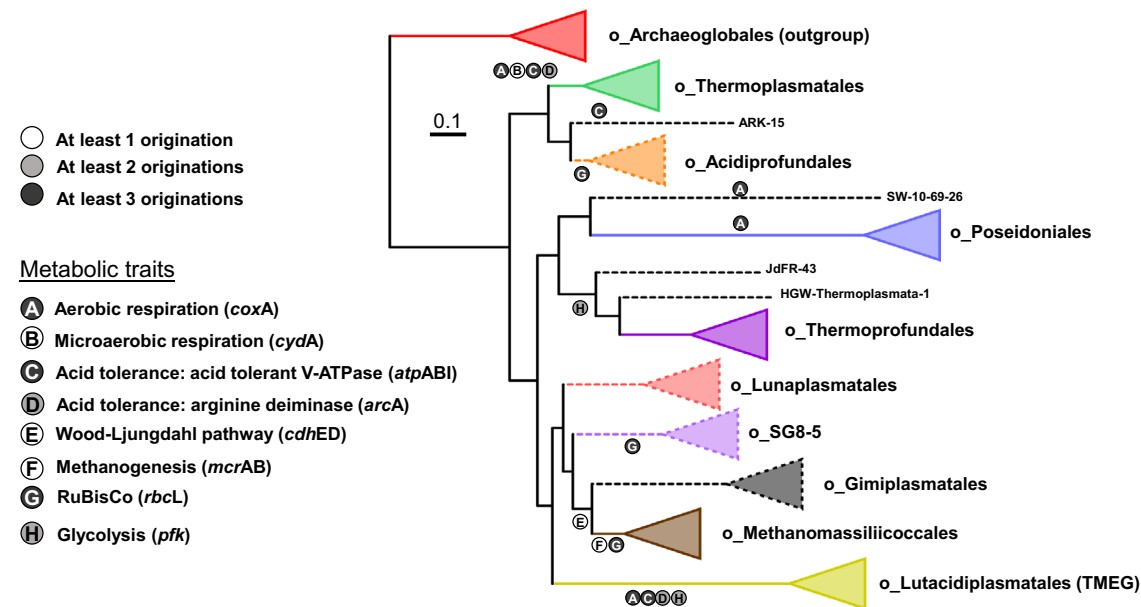

**Fig. 4 Evolution of Thermoplasmatota metabolisms.** The origins of key metabolisms were mapped on the phylogeny of Thermoplasmatota. Progressive gain and loss of genes between ancestral reconstructions and individual gene tree analysis were used to detect multiple independent lateral acquisitions of the same metabolisms into Thermoplasmatota lineages. The triangles represent collapsed clades. A dotted triangle indicates that this clade consists of less than four representative genomes, so originations may be underrepresented in the LCA of this clade (dotted branch). Marker genes of the selected metabolisms are in parenthesis. *cox*A (haem-copper oxygen reductase, subunit A; K02274), *cyd*A (cytochrome bd ubiquinol oxidase, subunit A; K00425), *atp*ABI (acid) (V/A-type atpase A, B and I subunits; K02117, K02118 and K02123), *arc*A (arginine deiminase; K01478), *cdh*DE (acetyl-CoA decarbonylase/synthase complex D; K00194 and E; K00197 subunits), *rbc*L (ribulose-bisphosphate carboxylase large chain; K01601) and *pfk* (ATP-dependent phosphofructokinase; K21071).

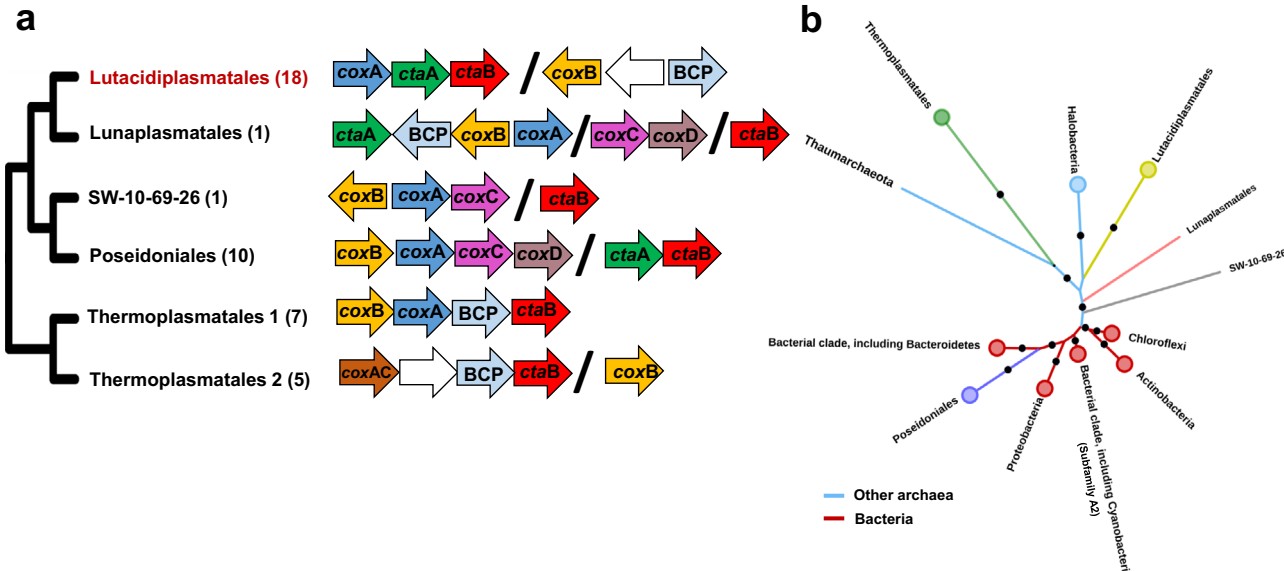

**Fig. 5 Complex IV genetic organisation and phylogeny. a** The physical genetic organisation of complex IV genes in Thermoplasmatota is conserved at the order level but highly variable at the phylum level. The complex IV loci in the genera *Ferroplasma*, *Acidiplasma* and *Picrophilus* (Thermoplasmatales 2), are different from those present in the rest of the Thermoplasmatales order (Thermoplasmatales 1). The black backslash (**/**) between genes indicate sections that are physically separated in the genome sequence, and white arrows represent hypothetical proteins. **b** The ML tree of the *cox*A subunit of haem-copper oxygen reductase shows that the Poseidoniales *cox*A gene clusters more closely with bacterial than other archaeal sequences, indicating an independent acquisition from a bacterial donor. Circles indicate collapsed clades. The Thaumarchaeaota, Lunaplasmatales and SW-10-69-26 branches are represented by the single genomes *Cenarchaeum symbiosum* A, Lunaplasmatales archaeon RBG-16-67-27 and Thermoplasmatota archaeon SW-10-69-26, respectively. Dots indicate branches with >75% UFBoot support. All analysed *cox*A genes are from A1 subfamily, apart from one bacterial clade belonging to A2 (labelled on the tree). The expanded *cox*A phylogenetic tree and trees of the genes *cox*B, *cta*A and *cta*B are presented in Supplementary Fig. 10–13.

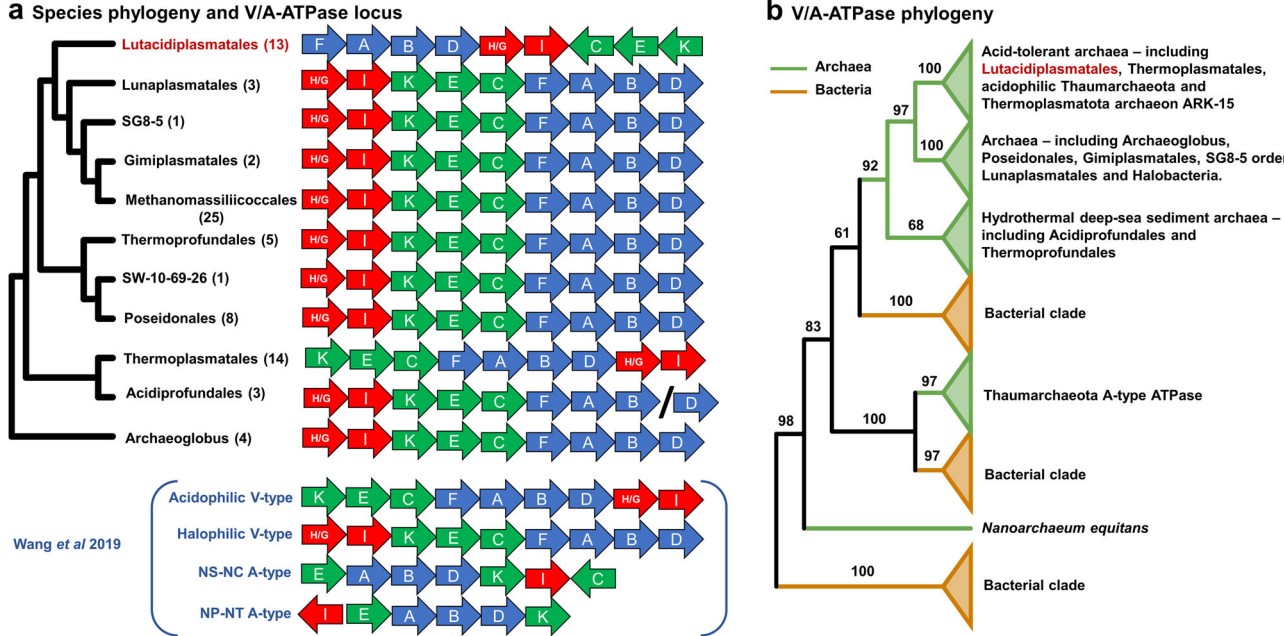

**Fig. 6 V/A-type ATPase locus gene order and phylogeny. a** The gene order of V/A-type ATPase genes is conserved at the order level and broadly conserved at the phylum level, except for Thermoplasmatales and Lutacidiplasmatales. The tree indicates the phylogeny of the species in which these loci are found. Four gene orders presented in Wang et al. 2019[29] are shown in parenthesis for comparison. The arrow colour indicates genes that tend to be adjacent in all gene orders and is added for comparison. **b** The ML tree of the three largest subunits (*atp*A, B and I) concatenated. Numbers on branches represent the percentage of UFBoot support. The tree was rooted by minimal ancestral deviation (MAD)[91]. The expanded *atp*ABI tree is presented in Supplementary Fig. 15.

gained twice independently, once into the Lutacidiplasmatales LCA and once into an ancestor pre-dating the Thermoprofundales LCA (Fig. 4). Additionally, *pfk* genes from Lutacidiplasmatales and Thermoprofundales were more closely related to bacterial homologues than each other, indicating two independent acquisitions into the phylum. A single acquisition of *pfk* into the phylum was statistically rejected by an AU test ($P = 0.02$) (Supplementary Fig. 23).

## Discussion

An expanded genome sampling of the Lutacidiplasmatales reveals that many group members are capable of aerobic respiration. This capacity also includes the previously analysed genome TMEG-bg1, which lacked complex IV but possessed a cytochrome bd ubiquinol oxidase (in our analysis). Several Lutacidiplasmatales also encode acetate and ethanol production genes, which indicate a facultative anaerobic strategy. This strategy is likely advantageous in the frequently waterlogged soils they inhabit, where molecular oxygen can be quickly depleted. Additionally, the presence of a putative sulfite oxidase capable of producing sulfate from sulfite and thiosulfate may explain their ecological distribution, such as in a water sample in which Lutacidiplasmatales predominate possessing 28–180 times higher sulfate (40 mM) than in instances where Lutacidiplasmatales was absent[45]. The prevalence of Lutacidiplasmatales in various acidic environments seems to be facilitated by several metabolic adaptations to low pH environments, such as an acid-tolerant ATPase and acidophilic peptidases (as noted in other acidophilic Thermoplasmatales species[25,29]), and a putatively acid tolerance-conferring arginine deiminase.

The evolutionary history of archaea is marked with several expansions into new environmental habitats and subsequent metabolic adaptation. That adaptation is likely driven by de novo gene origination, lateral acquisition of niche-relevant genes[46–48], and expansion of ancestral gene repertoires by duplication[34,49],

but the relative contributions of these processes across the archaeal tree, and the interplay between them, remain unclear. For example, it has been postulated that the evolutionary transition of the ammonia-oxidising Thaumarchaeota from hot springs to terrestrial environments co-occurred with the lateral acquisition of many new gene families and the subsequent duplication of these genes in specific terrestrial lineages[34]. Adam et al. noted that Thermoplasmatota represents a key model system for studying the processes underlying archaeal evolution and adaptation to contrasting environments due to the variety of habitats and metabolisms observed within this group[5]. Our study contributes to deciphering that complex history. Our analyses highlight the role of lateral gene transfer in driving metabolic transitions within Thermoplasmatota, including the convergent, independent acquisition of key genes from different (possibly bacterial) donors in different clades. For example, genes underpinning metabolisms such as oxidative phosphorylation, acid tolerance and heterotrophy were acquired multiple times from multiple donors. This suggests that Thermoplasmatota lineages have assimilated aspects of the indigenous microbial community's genetic repertoire to facilitate their niche adaptation following expansion into a new habitat. Additionally, some major metabolisms that appear conserved in divergent lineages of the phylum have likely arisen from convergent evolution rather than through vertical inheritance. In particular, Lutacidiplasmatales contributed to our understanding of Thermoplasmatales evolution by presenting independent acquisitions of several key metabolic traits. Evolutionary predictions are contingent on taxon sampling, and as more genomes become available our understanding of this important phylum will improve further. Due to the large variety of environments inhabited by the Thermoplasmatota, this group represents an exciting model for studying the gene content change mechanisms of habitat adaptation in archaeal lineages with complex evolutionary histories. It provides an excellent contrast to the more straightforward history of

environmental transition adaptation predicted for the ammonia-oxidising Thaumarchaeota[34,50,51].

Candidatus "Lutacidiplasma silvani" (sp. nov., gen. nov). "Luti" and "acidi" refer to this organism prevalence in acidic soil environments and "plasma" refers to its classification within the Thermoplasmatota. The name "'silvani" indicates that reference (type) genome, AcS3-62 (Genbank accession: GCA_022750295.1), was sequenced from a (pine) forest soil. It encodes genes for aerobic respiration and likely uses organic substrates, such as carbohydrates, peptides and fatty acids for organoheterotrophic growth. It is currently not cultured and only known from environmental sequencing. Genomes possess a high GC content (around 69%) and genome size of around 2.1 Mb.

Description of Lutacidiplasmatales (ord. nov.). Description is the same as for the genus Lutacidiplasma. Suff. -ales, ending to denote order. Type genus Lutacidiplasma gen. nov.

## Methods

**Sampling, sequencing and genome assembly**. Soils samples were collected from 11 sites around Scotland (UK) (Supplementary Data 1), and the environmental DNA was extracted using Griffith's protocol[52]. DNA libraries were prepared using Illumina TruSeq DNA PCR-Free Library Prep Kit, and sequencing was performed on the Illumina NovaSeq S2 platform ($8.7 \times 10^{10}$ bp per sample on average, Supplementary Data 19).

Reads were filtered using the READ_QC module[53], and high quality reads for each metagenome were assembled using MEGAHIT[54] and aligned back to the assembled contigs using bwa-mem v0.7.17[55] Binning of resulting contigs was performed with MaxBin2[56] and metaBAT2[57], and the results were consolidated using the Bin_refinement module[53]. Completeness and contamination of bins were estimated with CheckM[58], and only bins with a completeness of >45% and contamination of <10% were retained for further analysis. Genome coverage was calculated using CoverM v0.6.1 (https://github.com/wwood/CoverM). Bins were initially characterised using the classify_wf function in GTDB-Tk v1.7.0[59] using the R202 GTDB release. Genomes classified as p__Thermoplasmatota were extracted for further analysis. Defaults settings were used for all software if not otherwise stated.

**Collection of public genomes**. Genome sequences classified p__Thermoplasmatota and designated as the species representative genome by GTDB were downloaded from Genbank (June 2020) (Supplementary Data 2). TMEG-bg1 was also downloaded from IMG (https://img.jgi.doe.gov/). Representative genomes for the Poseidoniales were chosen based on the phylogenetic diversity and genome quality described previously[12] and downloaded from Genbank. Only genomes with a completeness of >45% and contamination of <10% were retained.

**Genome properties**. All genomes were annotated using Prokka v1.14[60], and several genome characteristics were estimated, including GC content (using QUAST[61]), total predicted genomic size (measured genome size (using QUAST[61]) corrected by the completeness score) and predicted optimal growth temperature (based on a machine learning model Tome[62]). Optimal growth temperature in ancestors of extant Thermoplasmatota was inferred with RidgeRace[63] and using the Tome predictions as leaf values. Environmental source information and genome sequence type (i.e. culture, SAG, MAG, etc) were retrieved from NCBI or the associated published study. The 5S, 16S and 23S rRNA and tRNA genes were identified using Barrnap v0.9 (–kingdom arc, archaeal rRNA) (https://github.com/tseemann/barrnap) and tRNAscan-SE v2.0.5[64] (-A, archaeal tRNA), respectively. The 16S rRNA genes were compared using BLASTn v2.9.0[65].

**Marker gene selection and phylogenomic reconstruction**

*Datasets*. This study used three datasets to build a full Thermoplasmatota tree (124 genomes), a Thermoplasmatota tree containing only higher-quality genomes (>70% completeness, <5% contamination)(100 genomes), and a tree restricted to the Lutacidiplasmatales (40 genomes) (see detailed description of the genomes in SI Phylogenomics).

*Gene marker selection*. For each dataset, conserved single-copy marker genes were detected using Roary (-i 50, -iv 1.5)[66]. Single-copy marker genes were defined as those present in a single copy in each genome and present in at least 50% of the genomes for the full dataset and the Lutacidiplasmatales-specific dataset (Supplementary Data 20), or 70% of genomes in the higher-quality dataset. Single-copy marker genes were aligned individually using MAFFT L-INS-i[67], and spurious sequences and poorly aligned regions were removed with trimAl (automated 1, resoverlap 0.55 and seqoverlap 60)[68]. Alignments were removed from further analysis if they presented evidence of recombination using the PHItest[69]. The remaining single-copy marker gene alignments from each of the three datasets were concatenated into supermatrices.

*Phylogenomic tree reconstruction*. A phylogenomic tree of the Lutacidiplasmatales-specific dataset was constructed from the supermatrix with IQ-TREE 2.0.3[70], using the best fitting model in ModelFinder[71] for each alignment and an edge-linked partition model. Branch validation of the tree involved 1000 SH-aLRT test[72] and 2000 UFBoot replicates, and a hill-climbing nearest neighbour interchange (NNI) search was performed to reduce the risk of overestimating branch supports.

To establish a robust phylogeny of the Thermoplasmatota, we compared eight phylogenomic trees reconstructed using different approaches (full schematic workflow is provided in Supplementary Fig. 24). The first four trees were estimated by selecting gene markers from two taxonomic samplings (the full and higher-quality datasets). Maximum likelihood trees were constructed for each supermatrix of alignments with IQ-TREE 2.0.3[70], using the best fitting model in ModelFinder[71] for each alignment and an edge-linked partition model. The same two alignments were also subjected to an additional round of trimAl (automated1) on the supermatrix before using the mixture model LG + C60 + F, resulting in four species trees. Branch validation of each tree involved 1000 SH-aLRT test[72] and 2000 UFBoot replicates, and a hill-climbing nearest neighbour interchange (NNI) search was performed. A fifth tree was constructed using a concatenation of 17 ribosomal genes from the full dataset using the best fitting model in ModelFinder[71] for each alignment and an edge-linked partition model.

Single-gene trees were constructed for the 71 marker gene alignments from the higher-quality genome dataset using IQ-TREE 2.0.3[70] and the best fitting model in ModelFinder[71]. These single-gene trees were used to construct a supertree using the multispecies coalescence method implemented in ASTRAL v5.7.5[73]. A phylogenomic tree was also constructed for the full dataset concatenated supermatrix after SR4 recoding[74], using IQ-TREE 2.0.3[70] and the best fitting model in ModelFinder[71]. A final phylogenomic tree was constructed with the marker gene set from the higher-quality dataset, but constrain to the topology presented in Adam et al 2017[5], using IQ-TREE 2.0.3[70] and the mixture model LG + C60 + F.

*Topology testing in constraint trees*. The higher-quality dataset was also used to create ML trees with constrained topologies to either place Posedoniales and Thermoprofundales as basal paraphyletic groups of the Thermoplasmatota (as in Adam et al.[5]) or placing them as a basal monophyletic group (as suggested, albeit with poor support, by the SR4 recoded tree). The constrained and unconstrained trees were then compared with approximately unbiased test[75] and other statistical tests implemented in IQ-TREE 2.0.3[70] (-zb 10000 -zw -au) (Supplementary Fig. 4).

*Taxonomic and relative abundance affiliation*. Order and family level classifications were based on relative evolutionary divergence (RED)[76], and genus and species stratifications were defined by amino acid identity (AAI) amalgamative clustering with 70 and 95% thresholds for genus and species, respectively[77,78]. AAI clusters that were polyphyletic by the species tree topology were considered to be two separate genera. AAI between sets of genomes were calculated using CompareM (https://github.com/dparks1134/CompareM). Relative abundance of each genome within their metagenome was estimated by mapping each genome sequence to their original assembled metagenome using the Quant_bins module in metaWRAP[53].

**Predicting gene content changes across evolutionary history**. For the Thermoplasmatota dataset using only higher-quality genomes, gene families were detected with Roary[66] with low stringency (-i 35, –iv 1.3, –s), and sequences shorter than 30 amino acids and families with less than four sequences were removed from further analysis. All remaining sequences within each family were aligned using MAFFT L-INS-i[79], processed with trimAl (automated1)[68] and ML phylogenetic trees were constructed for each alignment as described above. The majority of the gene family trees (6050 of the 6059) could be probabilistically reconciled against the supermatrix tree using the ALEml_undated algorithm of the ALE package[80] to infer the numbers of duplications, intra-LGTs, losses and originations on each branch of the supermatrix species tree. The probable origination points were also predicted using these data (Supplementary Data 21). Genome incompleteness was probabilistically accounted for within ALE using the genome completeness values estimated by CheckM[58]. The mechanism of gene content change on every branch of the species tree was estimated using branchwise_numbers_of_events.py[34]. The number of intra-LGTs to and from every branch of the species tree was estimated with calc_from_to_T.sh (see https://github.com/SheridanPO/ALE_analysis). All phylogenomic trees were visualised using iTOL[81].

A punctuation score was given to each mechanism of gene content change to measure the extent to which the branches with the highest numbers of events influence the total number of events for a given genome dataset. For a given mechanism, the punctuation score is calculated by dividing the sum of event numbers in the 10% of branches with the highest event numbers by 10% of the sum of events on all branches.

$$\text{Punctuation score} = \sum\nolimits_{\text{events in top10\%branches}} / (\sum\nolimits_{\sum \text{events in all branches}} * 0.1)$$

Gene family originations were predicted to result from inter-phyla lateral gene transfer if the medoid of the gene family possesses homologues in a database of UniRef90[82] sequences with strain-level designations and excluding Thermoplasmatota matches. The origins of these putatively laterally acquired gene families were estimated using the best match (as determined by bit score) in this database.

Probabilistic ancestral genome reconstructions were created for each branch of the species tree using gene_copies_at_node.py[34], giving a list of gene families predicted to be present in each ancestor. When the KEGG ortholog numbers corresponding to these gene families are present in a parent but not in the descendant branch genome reconstruction, they are considered functional gene loss. In the reverse situation, they are regarded as functional gene gains. This method enabled the prediction of multiple independent gene family gains and was combined with individual gene tree analysis to infer whether the transfer was from outside of the phylum.

**Functional annotation of gene families**. For each protein family, a medoid sequence (the sequence with the shortest summed genetic distances to all sequences in the family) was calculated under the BLOSUM62 substitution matrix using DistanceCalculator in Phylo (https://biopython.org/wiki/Phylo). Medoids were annotated against the KEGG database[37] using GhostKOALA[83], against the arCOG database[36] using Diamond BLASTp[84] (best-hit and removing matches with e-value >10⁻⁵, % ID < 35, alignment length <80 or bit score <100) and against the Pfam[85] and TIGRFAM[86] databases using hmmsearch[87] (HMMER v3.2.1) (-T 80). Annotation of specific gene families of interest is described in SI: Functional annotation of gene families. The subfamily classification of *cyd*A was performed using hmmsearch (-T 80) with the *cyd*A subfamily database[21]. The subfamily classification of *cox*A genes was performed using the haem-copper oxygen reductase database[44].

Carbohydrate active enzymes were annotated using profile HMM from dbCAN (http://bcb.unl.edu/dbCAN2/) (filtered with hmmscan-parser.sh and by removing matches with mean posterior probability <0.7). Extracellular peptidases were initially annotated using Pfam profile HMMs corresponding to MEROPs families, as described by Tully et al.[12], to identify peptidases and then predict signal peptides' presence in these genes using Signalp 5.0[88] (-org arch, archaeal signal peptides).

The presence of motility genes in Thermoplasmatota was initially assessed by the presence of the conserved archaellum subunits C (arCOG05119), D/E (arCOG02964), F (arCOG01824), G (arCOG01822) and J (arCOG01809). However, Tully et al. indicated that Poseidoniales species might possess divergent motility loci[12]. Therefore, profile HMMs of archaeal flagellin (PF01917) and *flaH* (PF06745) genes were used as markers for possible divergent motility, even when other archaellum related genes were absent.

**Evolutionary history of selected gene families**. A selection of 204 phylogenetically diverse genome sequences (176 bacterial and 28 archaeal) was downloaded from GenBank (Supplementary Data 15). These genomes were annotated against the KEGG database[89] using GhostKOALA[83]. Protein sequences of genes annotated as particular K numbers of interest were extracted from the genomes and combined with corresponding protein sequences in the Thermoplasmatota genomes. Sequences less than 50 amino acids long were removed, and the remaining sequences were aligned using MAFFT L-INS-I[67] (auto-mated1) and sequences whose alignments were composed of >50% gaps were removed. A ML phylogenetic tree was constructed using IQ-TREE 2.0.3[70] with 1000 UFBoot replicates, an NNI search and the best substitution model selected by ModelFinder[71]. For the phylogeny of the V/A-ATPase, genes from *Nitrosocosmicus oleophilis* MY3, *Nitrosotalea okcheonensis* CS and *Nitrososphaera viennensis* EN76 were added to the analysis. Single acquisitions of genes into the phylum were assessed by comparing the unconstrained gene tree with a tree constrained to form a monophyletic Thermoplasmatota clade. These topologies were then compared with approximately unbiased tests and other statistical tests implemented in IQ-TREE 2.0.3[70] (-zb 10000 -zw -au).

**Notes**. Thermoplasmatales, Acidiprofundales, Thermoprofundales, Poseidoniales, Lunaplasmatales, Gimiplasmatales and Lutacidiplasmatales are not validly published names under the International Code of Nomenclature of Prokaryotes and thus can be considered as candidatus taxa. The *Candidatus* prefix was omitted from these taxa in the manuscript for brevity. Additional information regarding the gene family history of *cta*B (Supplementary Data 22), progressive gene content change from the Thermoplasmatales LCA to the Lutacidiplasmatales LCA (Supplementary Data 23), predicted origins of genes transferred into the Lutacidiplasmatales (Supplementary Data 24) and their loss and duplication in extant genomes (Supplementary Data 25), and extended information on phylogenomic trees used in this study (Supplementary Data 26) are presented in the Supplementary Information.

**Reporting summary**. Further information on research design is available in the Nature Research Reporting Summary linked to this article.

## Data availability

Accession numbers for the 36 newly sequenced genomes presented in this study can be found in Supplementary Data 2 and under the NCBI BioProject PRJNA795910. Public data is available from NCBI (www.ncbi.nlm.nih.gov), IMG (https://img.jgi.doe.gov/), KEGG (https://www.genome.jp/kegg/), dbCAN (http://bcb.unl.edu/dbCAN2/download/), arCOG (https://ftp.ncbi.nih.gov/pub/wolf/COGs/arCOG/), PFAM (https://pfam.xfam.org/), TIGRFAM (http://tigrfams.jcvi.org/cgi-bin/index.cgi) and GTDB R202 (https://data.gtdb.ecogenomic.org/releases/). Source data are provided with this paper.

## Code availability

Scripts for general manipulation of ALE outputs have been deposited at https://github.com/Tancata/phylo/tree/master/ALE (https://doi.org/10.5281/zenodo.4012549)[34], and additional scripts specific to this work have been deposited at https://github.com/SheridanPO/ALE_analysis (https://doi.org/10.5281/zenodo.6598435)[90].

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

## Acknowledgements

UKRI financially supported P.O.S. and Y.M. through the NERC grant (NE/R001529/1). In addition, C.G.-R. and T.A.W. were supported by a Royal Society University Research Fellowship (URF150571 and UF140626). We thank Tony Travis for his support with Biolinux. The authors would also like to acknowledge the support of the Maxwell computer cluster funded by the University of Aberdeen.

## Author contributions

P.O.S., T.A.W., and C.G.-R. designed the study and developed the theory. P.O.S. assembled the 36 new genomes. P.O.S. collected the samples, and Y.M. performed DNA extraction. P.O.S., T.A.W., and C.G.-R. interpreted the results and wrote the manuscript. All authors accepted the final version of the manuscript.

## Competing interests

The authors declare no competing interests.
