## [Peer Review File · Nature Communications]

Recovery of Lutacidiplasmatales archaeal order genomes suggests convergent evolution in ThermoplasmatotaReviewers' Comments:

Reviewer #1:

Remarks to the Author:

The manuscript by Sheridan et al. reports 35 new MAGs from the archaeal TMEG clade within the Thermoplasmata, which has been for long strongly under sampled. They propose that TMEG are a new order that is widely spread in acidic environments, which they name Lutiacidiplasmatales, and propose a type-strain and the full etymology, as it should. Then, they annotate their metabolic diversity. Finally, they take advantage of this increase in genome coverage to reconstruct genome dynamics in the whole for the Thermoplasmata, a phylum that presents a large variety of metabolic adaptations. Overall, this is a nice study that substantially increases knowledge on this interesting part of the archaeal tree.

My only point of concern is the genome dynamics analysis. This is the weakest part of the study I think, as it relies heavily on the underlying topology, but the authors do not describe in detail how they chose it over alternative ones (everything is in Sup Discussion, but even there it is not super clear). Moreover, it should be made clearer that the acquisition of these new genomes allowed to do this thorough analysis at the phylum level, and this link is not very clear now. Without a solid discussion of how the underlying tree was chosen and how alternative topologies affect the inferences, I do not think the paper is ready for publication, as all downstream discussion on the evolution of key metabolisms depends on it.

Below some more specific comments:

Lines 60-105. It's hard to read a whole first paragraph without a main figure. The authors could move Figure S5 here.

Lines 66. You can remove the end of the sentence "an order-level group of the Thermoplasmata." As this has already been said in the introduction.

Lines 162-173. The *dsrAB* story is interesting, but a bit confusing. I did not understand what is shown in Figure S2, probably a result extracted from the ancestral reconstructions, but because you have not yet talked about it at this stage, I think Figure S3 could be enough here. I am not sure how realistic is an HGT from Lucidi to Eukaryotes. At least the authors should mention an alternative scenario where the eukaryotic gene comes from Thaumarchaeota (TACK, the closest relatives of Eukaryotes) and that Lucidi acquired it from Thausms. If I am not wrong, there have been previous analyses highlighting a high rate of HGT between Thausms and Thermoplasmatales, and the authors could mention it here.

Lines 183-192: It looks a bit bizarre to have a very small paragraph here. It should be expanded with more information. As you mention, the tree of Thermoplasmata is not stable from the literature, and your results will be of interest to many people and should fully described. Most importantly, if you want the readers to trust your genome evolution reconstructions, which heavily rely on the underlying topology, you must be more convincing that your tree is the right one by bringing back in the main text a summary of what is written in SI (Briefly, that you selected xxx conserved markers, that you tested xxx topologies from the literature, that you chose xxx based on xxx).

From SI, I was very confused. Figure S1, S2 and S3 do not show the data mentioned.

The increased availability of genomes in recent years and the new genomes sequenced in this study allow revisiting deep evolutionary relationships within the Thermoplasmata. Seven phylogenomic trees were created to estimate the phylogeny of the Thermoplasmata using different approaches (how?). All seven species trees constructed in this work for Thermoplasmata differed to some extent from the topology presented in Adam et al. 2017 (a thorough investigation spanning multiple archaeal phyla) (Figure S2, Topology B) and some other works^{23, 24}. Six trees (Trees 1-6 from Figure S1) resolved Acidiprofundales and

Thermoplasmatales as a basal monophyletic group in the Thermoplasmatota (Figure S2, Topology A). In contrast, the remaining tree (Tree 6 from Figure S1) implied an internal branching of this group (albeit with very poor support) (Figure S3, Topology C). Constraint tree statistical analysis of these three differing topologies strongly favoured topology A and could statistically reject topology B (where is this shown?).

Trees with this topology were used in further evolutionary analysis. Marker gene information for the full dataset trees is provided in Supplementary Data20. Please give some more information, at least the number and how they were chosen. I could not find the number of markers nor the size of the supermatrices. Did you use a Thermoplasmatota core?

Line 193. I am missing a link here, please say that having obtained a resolved tree, you moved on to understand the dynamics in the whole phylum.

I would not begin the paragraph with the temperature or gene size, it looks a bit disconnected from the rest of the paragraph. Maybe this (and the paragraph on the calculation of the tree could be moved up together with Figure 2).

This paragraph should in fact start with line 206 and figure 4. I would move Figure 3 in sup mat.

Reviewer #2:

Remarks to the Author:

This paper assembled 35 archaeal MAGs from forest and grassland soil metagenomes, and the authors proposed a new name "Lutiacidiplasmatales" in the phylum Thermoplasmatota after extensive phylogenetic analysis. Meanwhile, they studied the evolutionary history of the phylum Thermoplasmatota by using a set of reference genomes from public databases. Aerobic respiration and acid tolerance were likely acquired independently by divergent lineages through convergent evolution rather than inherited from a common ancestor. Overall, this article provides insight into the metabolism and evolution of Lutiacidiplasmatales and Thermoplasmatota, but some issues need to be revised and clarified.

Main text:

Line 1: After reading the title and main text, it seems that this paper is mainly focused on reporting a new archaeal order called "Lutiacidiplasmatales". However, nearly half of the main text is talking about evolution of the phylum Thermoplasmatota. The reviewer is confused about the main theme of this paper, because evolution of the order Lutiacidiplasmatales and the phylum Thermoplasmatota are two independent topics. Although Lutiacidiplasmatales is an order in Thermoplasmatota, it is not located at the root of this phylum in the phylogenomic tree. As a result, this paper seems to be studying the evolution of Thermoplasmatota. However, the number and quality of genomes used in this study are far more enough to give robust conclusions in this study.

Line 52-55: How is this conclusion revealed? Please provide detailed information.

Line 64: The 35 genomes here is not clearly defined. Are they TMEG or novel lineages of terrestrial archaea?

Line 72: What does redundancy mean? How is this value calculated out?

Line 75-76: This sentence is irrelevant with the main conclusion of this study.

Line 101-105: It is not appropriate to put the description of Lutiacidiplasmatales here at the beginning of the results section. It is better suited at the end of the main text.

Line 113-116: Are there any references showing *cydA* and *cydB* are characteristic of microaerobic microorganisms?

Line 117-118: Which data supported this? Any phylogenetic trees?

Line 149-150: This sentence is misleading. Is the *arcA* in Lutiacidiplasmatales involved in acid tolerance in bacteria? Please provide evidence such as in vivo experiments.

Line 157-160: This sentence can be deleted because it is irrelevant with the acid tolerance of Lutiacidiplasmatales.

Line 171-173: This is too speculative.

Line 178-179: Did you compare the value of novelty between arCOG and other databases such as KEGG.

Line 206-207: What is the standard of selecting reference genomes and outgroup? The reviewer feels that the number and quality of genomes used in this study are far more enough to give robust conclusions in this study. In addition, all of the major lineages adjacent to Thermoplasmata should be included in the analysis of evolutionary history of Thermoplasmata, which may significantly impact the conclusion.

Line 403: The latest version of GTDB is R202, which is released a year ago. The reviewer suggests an update of the data used in the study.

Line 406: GEM catalog (10.1038/s41587-020-0718-6) should also be searched and analyzed because it contain huge MAGs which are not included in NCBI.

Line 532: The description of the NCBI BioProject PRJNA795910 should be released on the website at least during the reviewing process.

Line 728: Figure 1. The figure legend of this figure is too long. Please consider moving some information to the supplement.

Line 736: Why 70% AAI is used to determine a genera? Are there any references using this threshold?

Line 790: Figure 4. The number and quality of genomes used in this study are far more enough to give robust conclusions in this study.

Line 804: Figure 5. This figure can be moved to supplement since it is not discussed much and no important conclusion is about this figure in the main text.

Line 830: Please add some important information about the genomes in this study, such as GC content, numbers of 16S, 23S, and 5S rRNA genes, and accession numbers.

Supplementary Information:

Line 12: The 45% completeness used in the full dataset is a bit low. 50% is usually chosen as a minimum.

Line 16: 70% completeness is not regarded as high-quality. At least 80%-90% completeness would meet this.

Reviewer #3:
Remarks to the Author:

This is a high quality study that details an group (class) of archaea that have not received attention. TMEG as they are commonly known are with the Thermoplasmatota phylum. In this manuscript they did a detailed analyses of the metabolic capabilities and very nicely linked it to the evolutionary histories of this group. It is well written and the methods are state of the art. I personally would have liked to see a diagram of the metabolic pathways, but I would say its not really needed. All the important features are well described in the text.

In terms of its impact of the findings, I would say it not massive, however, given they are pretty broadly distributed and nature and they haven't been examined this carefully. A greater understanding of the ecology and evolution will be important in the future.

REVIEWER COMMENTS

The line numbers below refer to the track-change version of the manuscript.

Reviewer #1 (Remarks to the Author):

The manuscript by Sheridan et al. reports 35 new MAGs from the archaeal TMEG clade within the Thermoplasmata, which has been for long strongly under sampled. They propose that TMEG are a new order that is widely spread in acidic environments, which they name Lutiacidiplasmatales, and propose a type-strain and the full etymology, as it should. Then, they annotate their metabolic diversity. Finally, they take advantage of this increase in genome coverage to reconstruct genome dynamics in the whole for the Thermoplasmata, a phylum that presents a large variety of metabolic adaptations. Overall, this is a nice study that substantially increases knowledge on this interesting part of the archaeal tree.

We thank the reviewer for their positive appraisal of this work.

My only point of concern is the genome dynamics analysis. This is the weakest part of the study I think, as it relies heavily on the underlying topology, but the authors do not describe in detail how they chose it over alternative ones (everything is in Sup Discussion, but even there it is not super clear) .

As detailed below, the phylogenomics section has now been expanded in the main text.

Moreover, it should be made clearer that the acquisition of these new genomes allowed to do this thorough analysis at the phylum level, and this link is not very clear now.

We have now clarified the importance of the expanded genome sampling for our analyses by adding text to the introduction (ln 57-59):

“The addition of Lutiacidiplasmatales to the analysis of Thermoplasmatota revealed the convergent evolution of genes involved in aerobic respiration, acid tolerance and glycolysis”

And in the discussion (ln 421-423):

“In particular, Lutiacidiplasmatales contributed to our understanding of Thermoplasmatales evolution by presenting independent acquisitions of several key metabolic traits”

Without a solid discussion of how the underlying tree was chosen and how alternative topologies affect the inferences, I do not think the paper is ready for publication, as all downstream discussion on the evolution of key metabolisms depends on it.

We hope that our revision now includes sufficient information. Specific changes detailed below.

Below some more specific comments:

Lines 60-105. It's hard to read a whole first paragraph without a main figure. The authors could move Figure S5 here.

This section has now been more clearly linked to Table 1, which has been further improved. Unfortunately, Fig S5 does not provide results of enough significant value to be included as a main figure.

Lines 66. You can remove the end of the sentence “an order-level group of the Thermoplasmata.” As this has already been said in the introduction.

This has been removed.

Lines 162-173. The *dsrAB* story is interesting, but a bit confusing. I did not understand what is shown in Figure S2, probably a result extracted from the ancestral reconstructions, but because you have not yet talked about it at this stage, I think Figure S3 could be enough here.

The figure S2 is required to determine the most likely root of the tree. We have described this in more detail in the main text (ln 177-179).

“A phylogenetic tree of these genes in various prokaryotes and eukaryotes was reconstructed and was rooted at the position with the lowest ancestor deviation (see SI MAD rooting; Figure S2).”

I am not sure how realistic is an HGT from Lucidi to Eukaryotes. At least the authors should mention an alternative scenario where the eukaryotic gene comes from Thaumarchaeota (TACK, the closest relatives of Eukaryotes) and that Lucidi acquired it from Thausms. If I am not wrong, there have been previous analyses highlighting a high rate of HGT between Thausms and Thermoplasmatales, and the authors could mention it here.

We have now fully removed our hypothesis concerning the HGT based on another reviewer's comments.

Lines 183-192: It looks a bit bizarre to have a very small paragraph here. It should be expanded with more information. As you mention, the tree of Thermoplasmatota is not stable from the literature, and your results will be of interest to many people and should fully described. Most importantly, if you want the readers to trust your genome evolution reconstructions, which heavily rely on the underlying topology, you must be more convincing that your tree is the right one by bringing back in the main text a summary of what is written in SI (Briefly, that you selected xxx conserved markers, that you tested xxx topologies from the literature, that you chose xxx based on xxx).

The phylogenomics section has been expanded in the main text and linking to supplementary information has been improved (ln 201-211). We have also added an extra supp table with information on the marker gene number, supermatrix length and other information for each of the trees (Supplementary Data 26).

“To investigate their evolutionary history, we performed a range of phylogenomic analyses using different taxon sets (an expanded 124 genome set including all 35 new Lutiacidiplasmatales genomes, and a 100-genome subsample of the highest-quality genomes including 21 new genomes). We used two marker gene sets (conserved single-copy marker genes and a 17-gene ribosomal protein set) and four analytical approaches (maximum-likelihood trees inferred from marker gene concatenated using per-gene partitioning and the best-fitting mixture model, LG+C60+F; data recoding; supertree inference using ASTRAL on a set of 71 single marker gene trees). More details are provided in SI Phylogenomics Methods and Results, Figure S4; Supplementary Data 12)”

From SI, I was very confused. Figure S1, S2 and S3 do not show the data mentioned.

We thank the reviewer for pointing out these typos in the figure label numbers. This has now been corrected (SI: ln 96-100).

The increased availability of genomes in recent years and the new genomes sequenced in this study allow revisiting deep evolutionary relationships within the Thermoplasmatota. Seven phylogenomic trees were created to estimate the phylogeny of the Thermoplasmatota using different approaches (how?).

As described above, more information has now been provided in the main text (ln 201-211). A schematic workflow of this process is also provided in Figure S24.

All seven species trees constructed in this work for Thermoplasmatota differed to some extent from the topology presented in Adam et al. 2017 (a thorough investigation spanning multiple archaeal

phyla) (Figure S2, Topology B) and some other works^{23, 24}. Six trees (Trees 1-6 from Figure S1) resolved Acidiprofundales and Thermoplasmatales as a basal monophyletic group in the Thermoplasmatota (Figure S2, Topology A). In contrast, the remaining tree (Tree 6 from Figure S1) implied an internal branching of this group (albeit with very poor support) (Figure S3, Topology C). Constraint tree statistical analysis of these three differing topologies strongly favoured topology A and could statistically reject topology B (where is this shown?).

The information for AU testing is shown in Figure S4 and the results of other statistical tests are presented in Data 12. As described above, we have now linked the supplementary phylogenomic information more explicitly to the main text (In 208-210).

Trees with this topology were used in further evolutionary analysis. Marker gene information for the full dataset trees is provided in Data 20. Please give some more information, at least the number and how they were chosen. I could not find the number of markers nor the size of the supermatrices. We have created a new table (Supplementary Data 26) that provides information on the gene marker number, supermatrix length and other information for each of the trees.

Did you use a Thermoplasmatota core?

Yes, we used three marker gene sets: conserved single-copy genes from the expanded genome selection (108 marker genes), conserved single-copy genes from the highest-quality genome selection (71 marker genes) and a 17-gene ribosomal protein set. We have now explicitly linked the detailed description of gene marker selection to the main text (In 208-210).

Line 193. I am missing a link here, please say that having obtained a resolved tree, you moved on to understand the dynamics in the whole phylum.

A linking sentence has now been added (In 214-215):

“The resolved phylogenomic tree allowed investigation of the genome dynamics in the whole phylum.”

I would not begin the paragraph with the temperature or gene size, it looks a bit disconnected from the rest of the paragraph. Maybe this (and the paragraph on the calculation of the tree could be moved up together with Figure 2). This paragraph should in fact start with line 206 and figure 4. Temperature and genome size sections have been moved to the previous paragraph (In 215-227) which has been retitled “Phylogenomic tree and genome characteristics of Thermoplasmatota” (In 196).

I would move Figure 3 in sup mat.

We prefer to keep figure 3 in the main text as it is discussed at length in the main text (In 244-274) and these results represent key findings of our study. In particular, it summarises the phylogenetic positioning of the number of events for each investigated mechanism of gene content change.

Reviewer #2 (Remarks to the Author):

This paper assembled 35 archaeal MAGs from forest and grassland soil metagenomes, and the authors proposed a new name "Lutiacidiplasmatales" in the phylum Thermoplasmatota after extensive phylogenetic analysis. Meanwhile, they studied the evolutionary history of the phylum Thermoplasmatota by using a set of reference genomes from public databases. Aerobic respiration and acid tolerance were likely acquired independently by divergent lineages through convergent evolution rather than inherited from a common ancestor. Overall, this article provides insight into the metabolism and evolution of Lutiacidiplasmatales and Thermoplasmatota, but some issues

needs to be revised and clarified.

We thank the reviewer for their detailed and supportive review.

Main text:

Line 1: After reading the title and main text, it seems that this paper is mainly focused on reporting a new archaeal order called "Lutiacidiplasmatales". However, nearly half of the main text is talking about evolution of the phylum Thermoplasmatota. The reviewer is confused about the main theme of this paper, because evolution of the order Lutiacidiplasmatales and the phylum Thermoplasmatota are two independent topics. Although Lutiacidiplasmatales is an order in Thermoplasmatota, it is not located at the root of this phylum in the phylogenomic tree. As a result, this paper seems to be studying the evolution of Thermoplasmatota.

The title "The new archaeal order Lutiacidiplasmatales reveals convergent evolution in Thermoplasmatota" does not only suggest a reporting of a new order but clearly indicate the use of those new genomes to investigate the mechanisms of genome evolution in the whole phylum.

However, the number and quality of genomes used in this study are far more enough to give robust conclusions in this study.

The quality and quantity of the genomes used in this study are sufficient to validate our conclusions, including on the existence of convergent evolution in Thermoplasmatota.

Quantity: Gene tree-species tree reconciliation requires considerable computational resources, with the formation of 1000's of gene trees and then performing 1000's of reconciliation. Here, we selected the genomes to be used by including all Lutiacidiplasmatales genomes available at the time and a single species level representative for the other orders of the phylum. These results based on these selected genomes were further validated by inference and inspection of single gene trees for the key genes on a more extensive number of organisms, including an expanded sampling of archaeal and bacterial homologues. This two-step approach was able to identify convergent evolution of several key metabolisms in Thermoplasmatota and was able to statistically reject single acquisitions of these genes into the phylum, thus robustly supporting our conclusions.

Quality: The quality of genomes used in this study is adequate. The reconciliation technique is inherently forgiving of genome incompleteness because genome completeness (estimated using CheckM and provided by the user as an input to the analysis) is accounted for in the estimation of the rates of duplication, transfer, and loss. In addition, it is worth pointing out that if gene absences are mis-interpreted as losses, the result will be an increase in the number of losses on terminal branches of the tree, with little impact on inferences deeper in the tree. Therefore, the 70% completeness minimum we set in this analysis is sufficient to give robust results.

Line 52-55: How is this conclusion revealed? Please provide detailed information.

We have added text to state the methods on which the conclusions are based (In 51-54) i.e. genome analysis, gene tree-species tree reconciliation and single gene tree analysis.

"In contrast to hypotheses based on analysis of the first available genome for the group, these organisms are predicted by **genome analysis** to be capable of aerobic respiration and oxidise, rather than reduce, sulfite generated from thiosulfate"

"Additionally, we reveal **by gene tree-species tree reconciliation and single gene tree analysis** that essential metabolic genes, such as those involved in aerobic respiration..."

Line 64: The 35 genomes here is not clearly defined. Are they TMEG or novel lineages of terrestrial archaea?

We modified the sentence as follows (ln 66-68): “Reconstruction of metagenome-resolved genomes recovered 35 genomes related to the Terrestrial Miscellaneous Euryarchaeota Group (TMEG), based on GTDB relative evolutionary divergence scores”

Line 72: What does redundancy mean? How is this value calculated out?

By “redundancy”, we meant the value labelled contamination in CheckM. We have changed “redundancy” to “contamination” throughout, as this is how CheckM describes it and how we describe it in the Methods section.

Line 75-76: This sentence is irrelevant with the main conclusion of this study.

As we have released this newly assembled genome within the study, it seems appropriate to mention it briefly, even though we do not examine it extensively.

Line 101-105: It is not appropriate to put the description of Lutiacidiplasmatales here at the beginning of the results section. It is better suited at the end of the main text.

We have rephrased this text (ln 105-107) and moved the full descriptions to the end of the Discussion section (ln 430-439).

Line 113-116: Are there any references showing *cydA* and *cydB* are characteristic of microaerobic microorganisms?

References showing the role of *cydAB* in microaerobic respiration have now been added (ln 122).

“18. Rice, C. W. & Hempfling, W. P. Oxygen-limited continuous culture and respiratory energy conservation in *Escherichia coli*. *J. Bacteriol.* 134, 115-124 (1978).

19. Tseng, C., Albrecht, J. & Gunsalus, R. P. Effect of microaerophilic cell growth conditions on expression of the aerobic (*cyoABCDE* and *cydAB*) and anaerobic (*narGHJI*, *frdABCD*, and *dmsABC*) respiratory pathway genes in *Escherichia coli*. *J. Bacteriol.* 178, 1094-1098 (1996).

20. Baughn, A. D. & Malamy, M. H. The strict anaerobe *Bacteroides fragilis* grows in and benefits from nanomolar concentrations of oxygen. *Nature* 427, 441-444 (2004).”

Line 117-118: Which data supported this? Any phylogenetic trees?

The *cydA* subfamilies were assigned based on the *cydA* subfamily database created by Murali et al. 2021. This has been explained more explicitly in the results (ln 124-125):

“The cytochrome bd ubiquinol oxidases identified in this study are members of the rarer quinol:O₂ oxidoreductase, qOR3 family, based on the *cydA* subfamily database²¹”

This was previously described in the SI Methods, but has now been moved to the main text methods to improve clarity (ln 542-543).

“The subfamily classification of *cydA* was performed using *hmmsearch* (-T 80) with the *cydA* subfamily database.”

Line 149-150: This sentence is misleading. Is the *arcA* in Lutiacidiplasmatales involved in acid tolerance in bacteria? Please provide evidence such as in vivo experiments.

We have rephrased the sentence to remove ambiguity (ln 157-158): “They encode an *arcA* arginine deiminase, which was demonstrated as an important gene for acid tolerance in other organisms²⁶⁻²⁸.”

Line 157-160: This sentence can be deleted because it is irrelevant with the acid tolerance of Lutiacidiplasmatales.

We have rephrased the sentence (Ln 166-167): “Lutiacidiplasmatales genomes also encode other stress resistance genes such as Uvr excinuclease,...”

Line 171-173: This is too speculative.

This has now been removed.

Line 178-179: Did you compare the value of novelty between arCOG and other databases such as KEGG.

The novelty vs KEGG for all genomes is presented in the Data 2 Supp. File. In addition, novelty vs KEGG in the example genome has been added to the main text (Ln 192-193) “and 58% were assigned to ortholog groups in the KEGG database”.

Line 206-207: What is the standard of selecting reference genomes and outgroup? The reviewer feels that the number and quality of genomes used in this study are far more enough to give robust conclusions in this study.

There is no standard for selecting reference genomes and outgroup and it depends on the research question. As detailed above, we are confident that the quantity and quality of the genomes used in our study are sufficient for inferring our conclusions, particularly considering the validation of findings with an expanded dataset for key metabolic genes.

In addition, all of the major lineages adjacent to Thermoplasmatota should be included in the analysis of evolutionary history of Thermoplasmatota, which may significantly impact the conclusion. The use of more outgroups would be necessary in work focused on the pre-Thermoplasmatota evolution, but this was not the focus of this study and so only the Archaeoglobales were included – as these were used to root the Thermoplasmatota species tree. We also performed single gene tree analyses including an expanded sampling of archaeal and bacterial homologues and these analyses could statistically reject single acquisitions into the phylum for the examined traits. Several of these independent acquisitions are predicted to be interdomain transfers, so the addition of other archaeal phyla is very unlikely to change the conclusions.

Line 403: The latest version of GTDB is R202, which is released a year ago. The reviewer suggests an update of the data used in the study.

The GTDB classification was reperformed with the newest version of GTDB-Tk v 1.7.0 using R202 as the database. The genome UBA184 is still the closest relative in GTDB to all of the Lutiacidiplasmatales genomes.

Line 406: GEM catalog (10.1038/s41587-020-0718-6) should also be searched and analyzed because it contain huge MAGs which are not included in NCBI.

The GEM catalog is an interesting resource, which is sure to be valuable for future analyses. We decided not to search that database for extra MAGs to include in the present work, as our genome database was assembled before publication of this resource. New genomes are published continually at a high rate; adding new genomes would require all of our analyses to be re-run, taking several months, by which time more genomes would be available. As described above, we do not think the addition of extra genomes is likely to change our main biological conclusions, particularly given our two-step approach in which predictions of gene acquisition generated systematically were then evaluated with manually-curated single gene trees, making use of expanded taxon sampling.

Line 532: The description of the NCBI BioProject PRJNA795910 should be released on the website at least during the reviewing process.

The NCBI BioProject is now released and publicly available.

Line 728: Figure 1. The figure legend of this figure is too long. Please consider moving some information to the supplement.

Most of the length of this figure legend is used in giving the full names of the presented genes. Having these names in the legend is easier for the reader than referring to the SI, so we prefer to retain them. The figure legend is also within the editorial limits of this journal.

Line 736: Why 70% AAI is used to determine a genera? Are there any references using this threshold?

We use 70% AAI for genus because Rodriguez et al 2014 found that most members of the same genus had an AAI of around 70% or more. We choose 70% as a cutoff as a conservative value for placing genomes into the same genera. We have included references to papers where the link between AAI and taxonomic stratification is examined (In 493).

“69. Rodriguez-R, L. M. & Konstantinidis, K. T. Bypassing cultivation to identify bacterial species. *Microbe* 9, 111-118 (2014).

70. Konstantinidis, K. T. & Tiedje, J. M. Towards a genome-based taxonomy for prokaryotes. *J. Bacteriol.* 187, 6258-6264 (2005).”

Line 790: Figure 4. The number and quality of genomes used in this study are far more enough to give robust conclusions in this study.

We have commented on this point above. However, we have added a sentence in the Discussion explaining that as more genomes become available, our knowledge of this group will improve. “Evolutionary predictions are contingent on taxon sampling, and as more genomes become available our understanding of this important phylum will improve further”

Line 804: Figure 5. This figure can be moved to supplement since it is not discussed much and no important conclusion is about this figure in the main text.

We would prefer to keep Figure 5 in the main text as it helps the reader to understand the evolutionary history of complex IV, which we describe at length in In 288-316. The gene order plots suggest that the individual genes may have different evolutionary histories, indicating that concatenation of different complex IV genes would be inappropriate, and the *coxA* gene tree allows the reader to see the inter-phyla divergence in the complex IV, without needing to refer to the SI. This provides the reader with a clear example of multiple acquisitions of the key trait, aerobic respiration.

Line 830: Please add some important information about the genomes in this study, such as GC content, numbers of 16S, 23S, and 5S rRNA genes, and accession numbers.

These features have now been added to Table 1 and some less important features have been moved to Supplementary Data 2, which is now referenced in the Table 1 legend (In 898-900).

Supplementary Information:

Line 12: The 45% completeness used in the full dataset is a bit low. 50% is usually chosen as a minimum.

There is no standard value for completeness threshold and previous studies have used 45% or less as a threshold (Ren et al 2019, Sheridan et al 2020). In addition, only 3 of the 35 newly released genomes are below 50% (48%, 46%, and 45%) and these 3 genomes were not used in the gene tree – species tree reconciliation.

Line 16: 70% completeness is not regarded as high-quality. At least 80%-90% completeness would meet this.

We have now changed “high-quality genomes” to “higher-quality genomes” throughout the text to avoid confusion with any specific “high-quality genome” definition, such as that of MIMAG and we have added a sentence in the main text (ln 243-244) to state explicitly what we term “higher-quality”.

“A selection of 96 higher-quality (>70% completeness, <5% contamination) Thermoplasmatota genomes...”

Reviewer #3 (Remarks to the Author):

This is a high quality study that details an group (class) of archaea that have not received attention. TMEG as they are commonly known are with the Thermoplasmatota phylum. In this manuscript they did a detailed analyses of the metabolic capabilities and very nicely linked it to the evolutionary histories of this group. It is well written and the methods are state of the art. I personally would have liked to see a diagram of the metabolic pathways, but I would say its not really needed. All the important features are well described in the text.

In terms of its impact of the findings, I would say it not massive, however, given they are pretty broadly distributed and nature and they haven't been examined this carefully. A greater understanding of the ecology and evolution will be important in the future.

We thank the reviewer for their positive appraisal of this work.

Reviewers' Comments:

Reviewer #1:

Remarks to the Author:

I am satisfied by the answers. I am still not convinced about the use of a consensus tree for the gain and loss analysis as this may not be the correct topology, maybe I did not understand well and so the authors could explain this better at lines 204-211. The description of the phylogenomic analysis in SI could still be improved. Other than that I think the paper has increased in clarity and can now be published.

Reviewer #2:

Remarks to the Author:

Many thanks to the authors for the extensive edits and detailed responses for my previous comments, this is very much appreciate. The edits have overall approved the manuscript and I only have a few remaining comments.

1. why some major lineages adjacent to Thermoplasmatota did not included in the analysis of evolutionary history of Thermoplasmatota? I still consider that it may significantly impact the conclusion although authors argue that they have performed single gene tree analyses to prove. However, they did not provide the results to in the revised manuscript.

Reviewer #1 (Remarks to the Author):

I am satisfied by the answers. I am still not convinced about the use of a consensus tree for the gain and loss analysis as this may not be the correct topology, maybe I did not understand well and so the authors could explain this better at lines 204-211. The description of the phylogenomic analysis in SI could still be improved.

We have substantially worked on this section to improve its clarity, and this has been done in the main text and in the SI. In addition, we have moved some of the phylogenomic methods in the main text.

Other than that I think the paper has increased in clarity and can now be published.

Reviewer #2 (Remarks to the Author):

Many thanks to the authors for the extensive edits and detailed responses for my previous comments, this is very much appreciated. The edits have overall approved the manuscript and I only have a few remaining comments.

1. why some major lineages adjacent to Thermoplasmatota did not included in the analysis of evolutionary history of Thermoplasmatota? I still consider that it may significantly impact the conclusion although authors argue that they have performed single gene tree analyses to prove. However, they did not provide the results to in the revised manuscript.

We are unsure which “major lineages adjacent to Thermoplasmatota” the reviewer refers to. In our gene tree analyses, we have included genes from multiple adjacent archaeal lineages and bacterial lineages. These are presented in Figure 5B, Figure 6B, Figures S10-S13 and Figures S15-S23. The likelihood of single acquisition (monophyly) and multiple acquisitions (polyphyly) of selected genes were statistically compared and the results are presented in Supplementary Data 16.

The use of genes from other archaea and bacteria is explained at the beginning of the section, but we have now stated this throughout the section to increase clarity.

Ln 281 “*Targeted gene tree analysis, including genes from other archaea and bacteria, revealed*”

Ln 311 “*(subunits A, B and I), including genes from other archaea and bacteria, revealed*”

Ln 339 “*these genes, including genes from other archaea and bacteria (Figure S20, S21)*”

Additionally the size of most of the Supplementary Figures has been increased, making the text in the individual leaves of the trees more visible.